# Timeless couples G-quadruplex detection with processing by DDX11 helicase during DNA replication

Leticia K Lerner[1,†,‡], Sandro Holzer[2,†], Mairi L Kilkenny[2], Saša Šviković[1], Pierre Murat[1], Davide Schiavone[1], Cara B Eldridge[1], Alice Bittleston[2], Joseph D Maman[2], Dana Branzei[3] [ID], Katherine Stott[2], Luca Pellegrini[2,*] [ID] & Julian E Sale[1,**] [ID]

## Abstract

Regions of the genome with the potential to form secondary DNA structures pose a frequent and significant impediment to DNA replication and must be actively managed in order to preserve genetic and epigenetic integrity. How the replisome detects and responds to secondary structures is poorly understood. Here, we show that a core component of the fork protection complex in the eukaryotic replisome, Timeless, harbours in its C-terminal region a previously unappreciated DNA-binding domain that exhibits specific binding to G-quadruplex (G4) DNA structures. We show that this domain contributes to maintaining processive replication through G4-forming sequences, and exhibits partial redundancy with an adjacent PARP-binding domain. Further, this function of Timeless requires interaction with and activity of the helicase DDX11. Loss of both Timeless and DDX11 causes epigenetic instability at G4-forming sequences and DNA damage. Our findings indicate that Timeless contributes to the ability of the replisome to sense replication-hindering G4 formation and ensures the prompt resolution of these structures by DDX11 to maintain processive DNA synthesis.

**Keywords** DNA replication; fork protection complex; G-quadruplex; replisome; timeless
**Subject Categories** DNA Replication, Recombination & Repair; Structural Biology
**The EMBO Journal (2020) 39: e104185**

See also: **CH Freudenreich** (September 2020)

## Introduction

DNA can create significant impediments to its own replication through the formation of secondary structures. When unwound, certain sequences, often repetitive or of low complexity, can adopt a variety of non-B form structures, including hairpins, cruciforms, triplexes and quadruplexes (Mirkin & Mirkin, 2007). It is increasingly clear that secondary structure formation is a frequent event during replication, even at genomically abundant sequences previously thought not to be a major source of difficulty (Šviković et al, 2019). To prevent such sequences causing havoc with the genetic and epigenetic stability of the genome, cells deploy an intricate network of activities to counteract secondary structure formation and limit its effects. These activities include proteins that bind and destabilise DNA structures and specialised helicases that unwind them (Lerner & Sale, 2019). In addition, the repriming activity of PrimPol can be deployed to confine a structure into a minimal region of single-stranded DNA (ssDNA), limiting the potential dangers of exposing extensive ssDNA in a stalled replisome (Schiavone et al, 2016; Šviković et al, 2019).

G4s are one of the most intensively studied and potent structural replication impediments. G4s arise in consequence of the ability of guanine to form Hoogsteen base-paired quartets (Gellert et al, 1962). In favourable sequence contexts, comprising runs of dG separated by variable numbers of non-G bases, stacks of G quartets form G4 secondary structures. Current estimates suggest that over 700,000 sites in the human genome have the potential to form G4s (Chambers et al, 2015). While some of these G4s may have important roles in genome physiology, all pose a potential threat to DNA replication and sites with G4-forming potential have been linked to both genetic and epigenetic instability (Šviković & Sale, 2017; Kaushal & Freudenreich, 2019).

1   MRC Laboratory of Molecular Biology, Cambridge, UK
2   Department of Biochemistry, University of Cambridge, Cambridge, UK
3   IFOM, Fondazione Italiana per la Ricerca sul Cancro, Institute of Molecular Oncology, Milan, Italy
    *Corresponding author. Tel: +44 1223 760469; E-mail: lp212@cam.ac.uk
    **Corresponding author. Tel: +44 1223 267099; E-mail: jes@mrc-lmb.cam.ac.uk
    †These authors contributed equally to this work
    ‡Present address: Centre de Recherche des Cordeliers, Cell Death and Drug Resistance in Hematological Disorders Team, INSERM UMRS 1138, Sorbonne Université, Paris, France

Precisely, how DNA structures are detected and resolved by the replication machinery remains unclear. Many of the factors involved in processing G4 secondary structures, for instance FANCJ and REV1 (Kruisselbrink *et al*, 2008; London *et al*, 2008; Wu *et al*, 2008; Youds *et al*, 2008; Sarkies *et al*, 2010), do not appear to be constitutive components of the replisome (Dungrawala *et al*, 2015). It is thus likely that core components of the replisome will act as "first responders" to DNA structures and play an important role coupling their detection with suppressing their deleterious effects on DNA synthesis. Particularly interesting in this context is a subset of replisome components known as the fork protection complex (FPC). The FPC comprises four main proteins—Timeless, Tipin, Claspin and AND-1—that are conserved from yeast to mammals (Errico & Costanzo, 2012). FPC components associate with the replication fork via direct interactions with the CMG replicative helicase and replicative polymerases α, δ and ε (Nedelcheva *et al*, 2005; Numata *et al*, 2010; Cho *et al*, 2013; Bastia *et al*, 2016; Kilkenny *et al*, 2017; Baretić *et al*, 2020). They also interact with DNA both directly (Tanaka *et al*, 2010) and indirectly via replication protein A (RPA) (Witosch *et al*, 2014). These interactions allow the FPC to remain at the fork, which ensures a normal speed of DNA synthesis (Yeeles *et al*, 2017). Additionally, the FPC has a series of functions that promote normal replisome progression and fork integrity: it is essential to avoid uncoupling of pol ε from the replicative helicase and consequent formation of long stretches of ssDNA (Katou *et al*, 2003; Lou *et al*, 2008). It also has a conserved role in S-phase checkpoint activation in response to DNA damage, including checkpoint kinase activation, cell cycle arrest and maintenance of the integrity of the replication fork (Chou & Elledge, 2006; Gotter *et al*, 2007; Unsal-Kaçmaz *et al*, 2007; Yang *et al*, 2010). Furthermore, it plays an important, but incompletely understood, role in maintaining sister chromosome cohesion (Chan *et al*, 2003; Leman *et al*, 2010).

The FPC is thus well placed to play a role in the detection and metabolism of DNA secondary structures that could impede DNA synthesis. Indeed, deficiency of both TOF1 in yeast and Timeless in human cells leads to replication fork stalling, repeat instability and fragility at secondary structure-forming sequences (Voineagu *et al*, 2008, 2009; Leman *et al*, 2012; Liu *et al*, 2012b; Gellon *et al*, 2019), underscoring the potential importance of Timeless in maintaining processive replication through regions of the genome capable of forming secondary structures.

Although Timeless itself does not appear to possess catalytic activity that would process DNA secondary structures, it interacts with the DNA helicase DDX11 (Calì *et al*, 2016). DDX11 (or CHLR1), is a 5′–3′ Fe–S helicase of the same superfamily 2 as FANCJ, RTEL and XPD (Lerner & Sale, 2019). In humans, mutations in DDX11 cause Warsaw breakage syndrome, an extremely rare autosomal recessive disease characterised by microcephaly, growth retardation, cochlear abnormalities and abnormal skin pigmentation (Alkhunaizi *et al*, 2018). *In vitro*, DDX11 has unwinding activity on several non-duplex DNA structures, such as G4s (Wu *et al*, 2012a; Bharti *et al*, 2013), triplex DNA (Guo *et al*, 2015) and D-loops (Wu *et al*, 2012a). Further, the helicase activity of DDX11 is enhanced by Timeless (Calì *et al*, 2016). However, it remains unclear how Timeless and DDX11 collaborate *in vivo* in detecting and processing G4s during replication. Here, we provide *in vivo* evidence that Timeless and DDX11 operate together to ensure processive replication of G4-forming DNA. We report a previously unappreciated DNA-binding domain (DBD) in the C-terminus of Timeless, which exhibits specificity towards G4 structures. We propose that Timeless plays a role in the detection of G4 structures at the replication fork, recruiting DDX11 to unwind them and ensure processive replication is maintained, thereby avoiding G4-induced genetic and epigenetic instability.

# Results

## Timeless is required for processive replication of a genomic G4 motif

To address whether Timeless is involved in maintaining processive replication of G4-forming DNA *in vivo*, we disrupted the *TIMELESS* locus in chicken DT40 cells with CRISPR/Cas9-induced deletions. We isolated several *timeless* mutants with biallelic disruptions in exon 1, around the guide site (Appendix Fig S1). The *timeless* mutant cells were sensitive to cisplatin (Fig EV1), as previously observed in human cells depleted of Timeless (Liu *et al*, 2017). To assess the role of Timeless in the replication of a G4-forming sequence, we took advantage of the Bu-1 loss variant assay (Schiavone *et al*, 2014). The stable expression of the *BU-1* locus in DT40 is dependent on the maintenance of processive replication through a G4 motif located ~ 3.5 kb downstream of the promoter (Fig 1A). Prolonged pausing of leading-strand replication at this motif leads to loss of epigenetic information around the promoter of the gene and a permanent and heritable change in its expression (Sarkies *et al*, 2012; Schiavone *et al*, 2014, 2016; Guilbaud *et al*, 2017). This stochastic and replication-dependent generation of Bu-1 loss variants can be monitored by flow cytometry as *BU-1* encodes a surface glycoprotein. Small pools of Bu-1$^{high}$ wild-type and *timeless* cells were expanded in parallel for ~ 20 divisions (15–21 days), and the proportion of cells in each pool that had lost their Bu-1$^{high}$ status determined. We detected increased levels of Bu-1 expression instability in *timeless* DT40 cells compared to the wild-type cells, which retained their stable Bu-1$^{high}$ expression (Fig 1B and C). The instability of Bu-1 expression in *timeless* cells was fully reversed by expression of human Timeless (Fig 1C) and is dependent on the +3.5 G4 motif (Fig 1C). Cells deficient in Tipin (Abe *et al*, 2016), a constitutive interactor of Timeless within the FPC, also exhibit instability of *BU-1* expression (Fig 1D). These results show that Timeless is necessary to maintain processive DNA replication of a genomic G4 motif.

## Identification and characterisation of a Timeless DNA-binding domain

As a core component of the replisome, Timeless is intimately associated with DNA synthesis at the replication fork (Yeeles *et al*, 2017). *In vitro* data show that the Timeless–Tipin complex can bind to ssDNA through RPA (Witosch *et al*, 2014), and the Swi1-Swi3 complex, the fission yeast orthologue of Timeless–Tipin, was also shown to bind DNA (Tanaka *et al*, 2010). Inspection of the amino acid sequence of human Timeless revealed the presence of a conserved domain in its C-terminal half (residues 816–954; Fig 2A), with a predicted fold similarity to the myb-like proteins of the homeodomain-like superfamily, that bind double-stranded DNA (dsDNA)

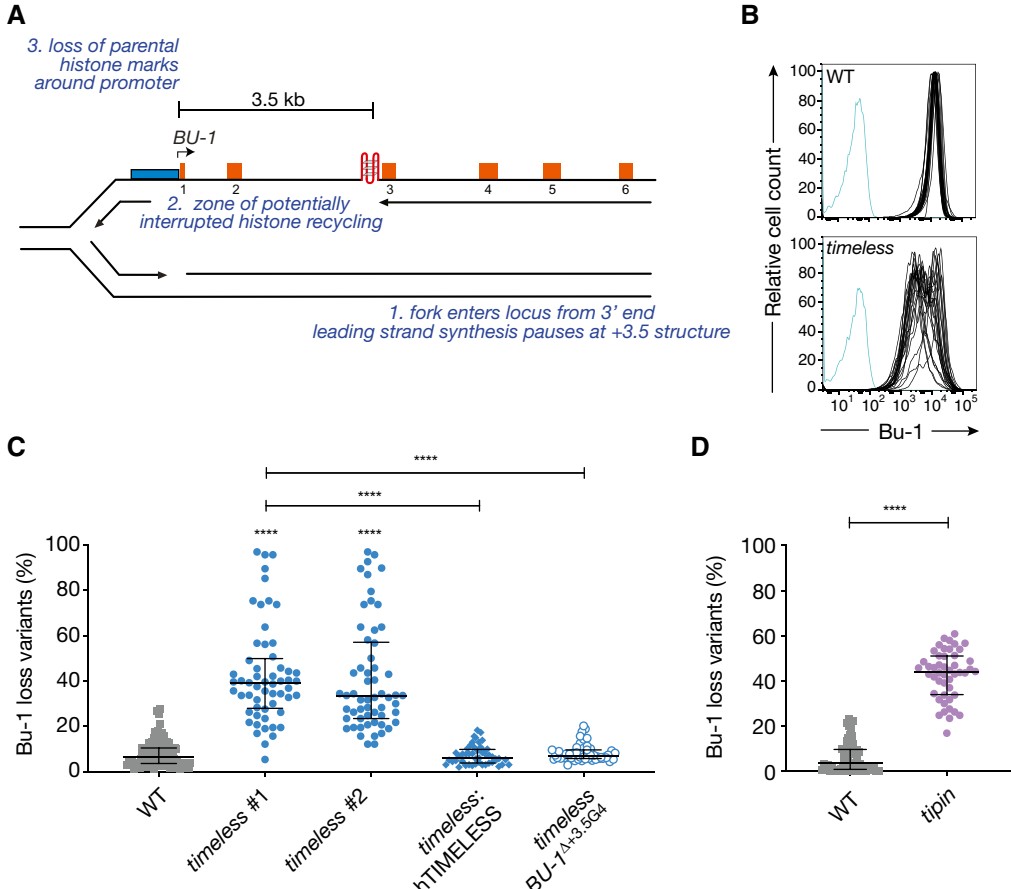

**Figure 1. Timeless and Tipin are required to maintain processive replication past G4 structures *in vivo*.**

A The *BU-1* locus as a model system to record G4-dependent replication stalling. The leading strand of a replication fork entering the locus from the 3′ end stochastically stalls at the +3.5 G4, leading to the formation of a region of ssDNA, with interruption of parental histone recycling and of histone modifications necessary to maintain normal expression of the locus (Schiavone *et al*, 2014).

B Instability of *BU-1* expression in *timeless* cells. FACS plots of wild-type and *timeless* (clone 1) DT40 cells stained with anti-Bu-1 conjugated with phycoerythrin. Each line represents the Bu-1 expression profile of an individual clonal population. Unstained controls are shown in blue.

C Fluctuation analysis for Bu-1 loss in wild-type DT40 cells and two independent *timeless* clones generated by CRISPR-Cas9 targeting (clones 1 and 2; Appendix Fig S1), *timeless* (clone 1) complemented by expression of human Timeless cDNA and a *timeless* mutant on a background in which the endogenous +3.5 G4 has been deleted (ΔG4) (Schiavone *et al*, 2014).

D Fluctuation analysis for Bu-1 loss in DT40 wild-type and *tipin* cells.

Data information: In (C) and (D), each symbol represents the percentage of cells in an individual clone expanded for 2–3 weeks that have lost Bu-1^high expression. At least two independent fluctuation analyses were performed, with 24–36 individual clones each cell line per repeat. Bars and whiskers represent median and interquartile range, respectively. ****$P < 0.0001$; one-way ANOVA.

with a tandem repeat of 3-helix bundles (named here N-term and C-term).

We used X-ray diffraction and NMR spectroscopy to investigate experimentally the structure and dynamics of the newly discovered Timeless domain. The 1.15 Å crystal structure of amino acids 885–947 (C-term), corresponding to a single myb-like fold, confirmed the presence of a three-helix bundle characteristic of the homeodomain superfamily of DNA-binding proteins, extended by the presence of a fourth C-terminal alpha helix unique to the Timeless domain (Fig 2B). The NMR structural ensemble of amino acids 816–954 revealed two well-converged domains (824–880 and 891–944, backbone r.m.s.d. 0.4 and 0.5 Å, respectively) connected by a linker (881–890) that was significantly less well converged, implying a high degree of flexibility between N- and C-term domains (Fig 2C).

This observation was confirmed by backbone dynamics measurements (Appendix Fig S2). The N- and C-terminal domains adopted the same three-dimensional fold; in particular, the N-terminal repeat shared the presence of an additional fourth helix, H4, as seen in the C-terminal repeat (Fig 2C).

In keeping with the similarity of its structure to known DNA binding domains (DBDs), we examined the ability of the Timeless domain to interact with DNA. We found that it bound with low micromolar affinity to both ss- and dsDNA probes (Fig 2D; top panel). A distinguishing feature of the Timeless DBD is the presence of a fourth alpha helix in both N- and C-terminal 3-helix bundle repeats. Superposition of the DBD C-term onto the structurally homologous domains of the telomeric protein TRF1 (Court *et al*, 2005) and bacterial cell cycle regulator GcrA (Wu *et al*, 2018) in

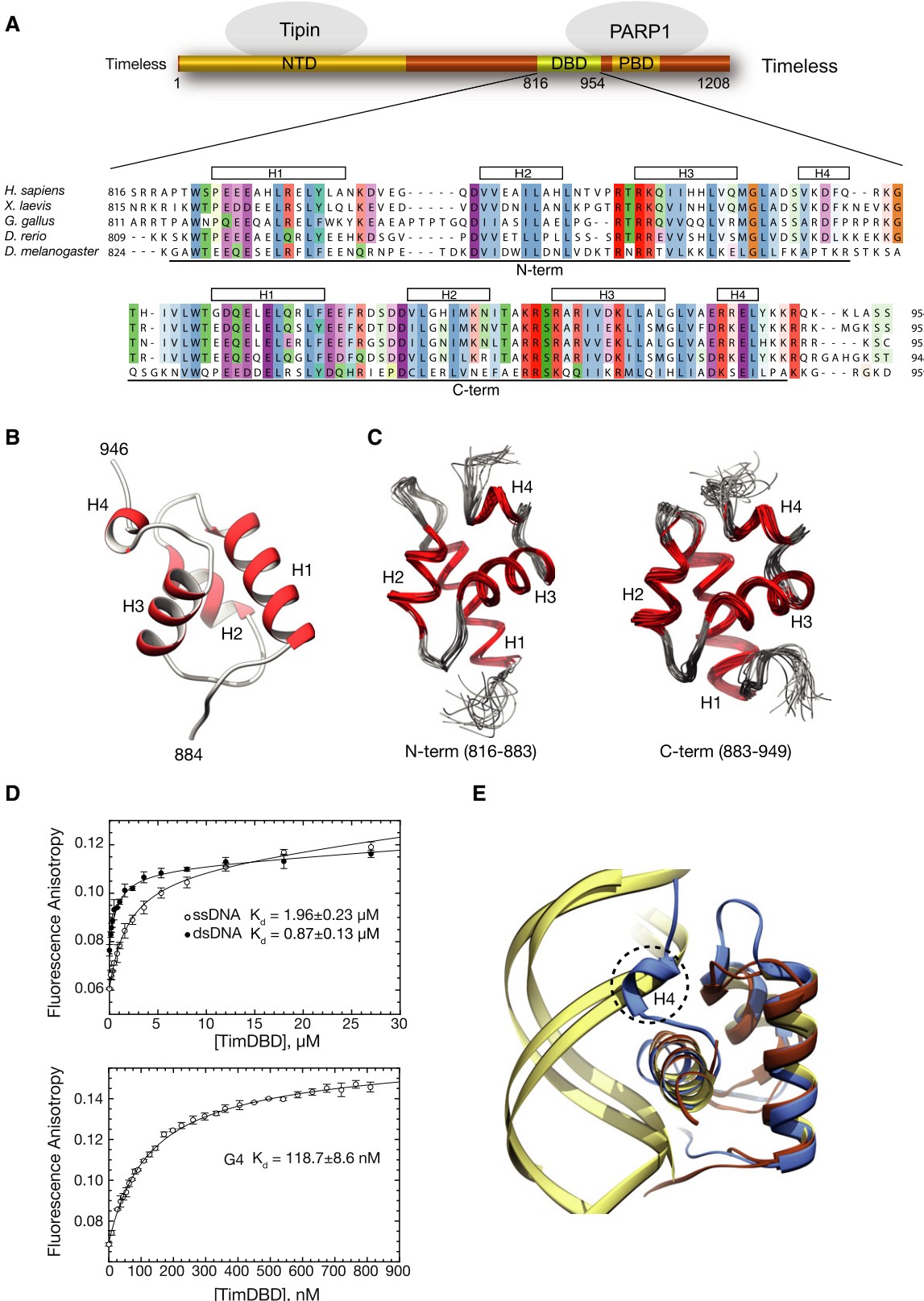

Figure 2.

◀

**Figure 2.   Identification and characterisation of a DNA-binding activity in Timeless.**

A   Schematic drawing of human Timeless and its known domain structure (NTD: N-terminal domain; DBD: DNA-binding domain; PBD: PARP-binding domain). A multiple sequence alignment of vertebrate Timeless sequences is shown underneath, with amino acid conservation coloured according to the Clustal colour scheme. The alignment is annotated with the extent and secondary structure elements of the two helical domains (N-term and C-term) composing the DBD.

B   Ribbon drawing of the 1.15 Å crystal structure at of the DBD C-term. Helices are in red and labelled H1–H4.

C   Ribbon drawings of the N-term and C-term domains of the DBD determined by NMR. The two domains are shown in the same orientatid to highlight their high degree of three-dimensional similarity. The superposition of the 20 lowest energy structures is shown for each domain.

D   The DNA-binding affinity of DBD was measured by fluorescence anisotropy, titrating the DBD protein against Cy3 3′-labelled ssDNA, dsDNA and G4 DNA (see Appendix Table S2 for sequence details). The top panel shows binding curves for ss- and dsDNA, and the bottom panel shows the binding curve for the G4 DNA substrate. The data points represent the mean of at least three independent experiments, and the error bars indicate one standard deviation (SD).

E   Ribbon diagram of the superposition of DBD C-term with the highly similar DNA-binding domains of telomeric protein TRF1 (PDB ID 1W0T) (Court *et al*, 2005) and the bacterial cell cycle regulator GcrA (PDB ID 5Z7I) (Wu *et al*, 2018) in complex with their DNA substrates. A similar DNA-binding mode by DBD would cause a steric overlap of helix H4 with the phosphate backbone of dsDNA. DBD C-term is in light blue, TRF1 and GcrA proteins in brown and their DNA substrates in khaki.

complex with their dsDNA substrates (Fig 2E) shows that, if the DBD were to adopt a similar mode of dsDNA binding, the fourth helix of both its N-term and C-term repeats would likely lead to a steric clash with the phosphate backbone of the DNA.

Although it is conceivable that the DBD might rearrange its conformation upon DNA binding, the fourth helix of both repeats participates in core hydrophobic interactions, mediated by conserved residues L872 and F879 (N-terminal repeat), and L936, V937 and L943 (C-terminal repeat), making such rearrangements unlikely. Given the functional context in which Timeless operates as a replisome component and the observations, presented in Fig 1, that it is required to maintain processive replication of the *BU-1* G4, we speculated that the C-terminal DNA-binding activity of Timeless might be directed towards recognition of DNA secondary structures that form transiently on the unwound template, such as G4s.

Indeed, when we tested a well-characterised G4 sequence present in the promoter of the *MYC* gene (Ambrus *et al*, 2005), the DBD bound to it with nanomolar affinity, and about one order of magnitude tighter than ds- or ssDNA (Fig 2D). This observation prompted us to ask whether the Timeless–Tipin complex, like the isolated DBD, is able to bind to the same G4 motif. To mimic the unwound template DNA, we embedded the G4 within a longer ssDNA sequence (ssG4; Appendix Table S2). We found that the Timeless–Tipin complex bound to ssG4 with low micromolar affinity and in a selective fashion, as it did not show measurable interactions with a hairpin DNA embedded within the same ssDNA (ssHP), or a mutated ssG4 sequence that had lost the ability to fold into a G4 (ss; Figs 3A and EV2, Appendix Table S2). We next tested a series of G4 sequences found in different genomic contexts and with different folding topologies: we found that the Timeless–Tipin complex bound to all of them, albeit with different affinities that varied several fold, whereas it did not show appreciable binding to ss- or dsDNA (Fig 3B).

These findings show that Timeless contains, in its C-terminal half, a previously unrecognised DBD, which closely resembles in structure the tandem repeat of three-helix bundles found in the homeodomain-like superfamily of transcription factors. While Timeless DBD binds to both ss- and dsDNA, it binds with ~ 10-fold greater affinity to a defined G4 DNA sequence. The preference for G4 DNA is retained by the Timeless–Tipin complex.

**The Timeless C-terminus is crucial for processive G4 replication *in vivo***

To further explore the *in vivo* contribution of the Timeless C-terminus to G4 replication, we generated a DT40 cell line expressing

a version of Timeless truncating the gene before the DBD, using CRISPR/Cas9 gene targeting to exon 16. This truncation also removes a domain previously reported to bind PARP1, the PARP1-binding domain (PBD) (Xie *et al*, 2015). This cell line exhibited instability of *BU-1* expression comparable to the *timeless* mutant (Fig 4) suggesting a role for the C-terminus of the protein in G4 replication. We confirmed this result by expressing human Timeless truncated at amino acid 816, and thus lacking both the DBD and PBD, in the *timeless* mutant (Appendix Fig S3). To further dissect this observation, we complemented *timeless* cells with truncated versions of human Timeless, lacking only the DBD (ΔDBD) or the PBD (PARP*). Timeless lacking either the DBD or the PBD largely restored, although not completely, the *BU-1* expression instability of the *timeless* mutant suggesting that the DBD and PBD act redundantly. Additionally, co-immunoprecipitation (Co-IP) experiments showed that neither region is required for binding the DDX11 helicase (Fig EV3), a known binding partner of Timeless and whose ability to unwind G4s is stimulated by Timeless (Calì *et al*, 2016). These results indicate that the C-terminus of the Timeless protein has an important role in G4 replication to which both the DBD and the PDB contribute, independently of DDX11 recruitment.

**DDX11 ensures processive G4 replication *in vivo***

Since Timeless itself lacks any catalytic activity, we next explored the extent to which the genetic interaction between Timeless and the DDX11 helicase accounted for G4 processing in this system. As noted above, Timeless interacts with DDX11 (Leman *et al*, 2010; Calì *et al*, 2016; Cortone *et al*, 2018), and this interaction has been shown to be important for sister chromatid cohesion (Cortone *et al*, 2018) and preservation of fork progression in perturbed conditions (Calì *et al*, 2016). Further, *in vitro*, DDX11 can unwind several DNA structures including G4s (Wu *et al*, 2012a; Bharti *et al*, 2013; Guo *et al*, 2015) and the activity of DDX11 is stimulated by Timeless (Calì *et al*, 2016). Recent work has shown that DDX11 interacts with Timeless through a domain encoded in exon 4 (Cortone *et al*, 2018). Although DDX11 was not identified by iPOND as constitutively present at normal or HU-stressed replication forks in human cells (Dungrawala *et al*, 2015), it interacts with several components of the core replisome in addition to Timeless, such as AND-1, POLD1 (Simon *et al*, 2020) and PCNA (Farina *et al*, 2008). We examined whether, in DT40, DDX11 is able to associate with chromatin-bound PCNA and whether this association is increased following exposure to the G4 ligand pyridostatin (PDS). We therefore performed chromatin immunoprecipitation with PCNA and blotted for

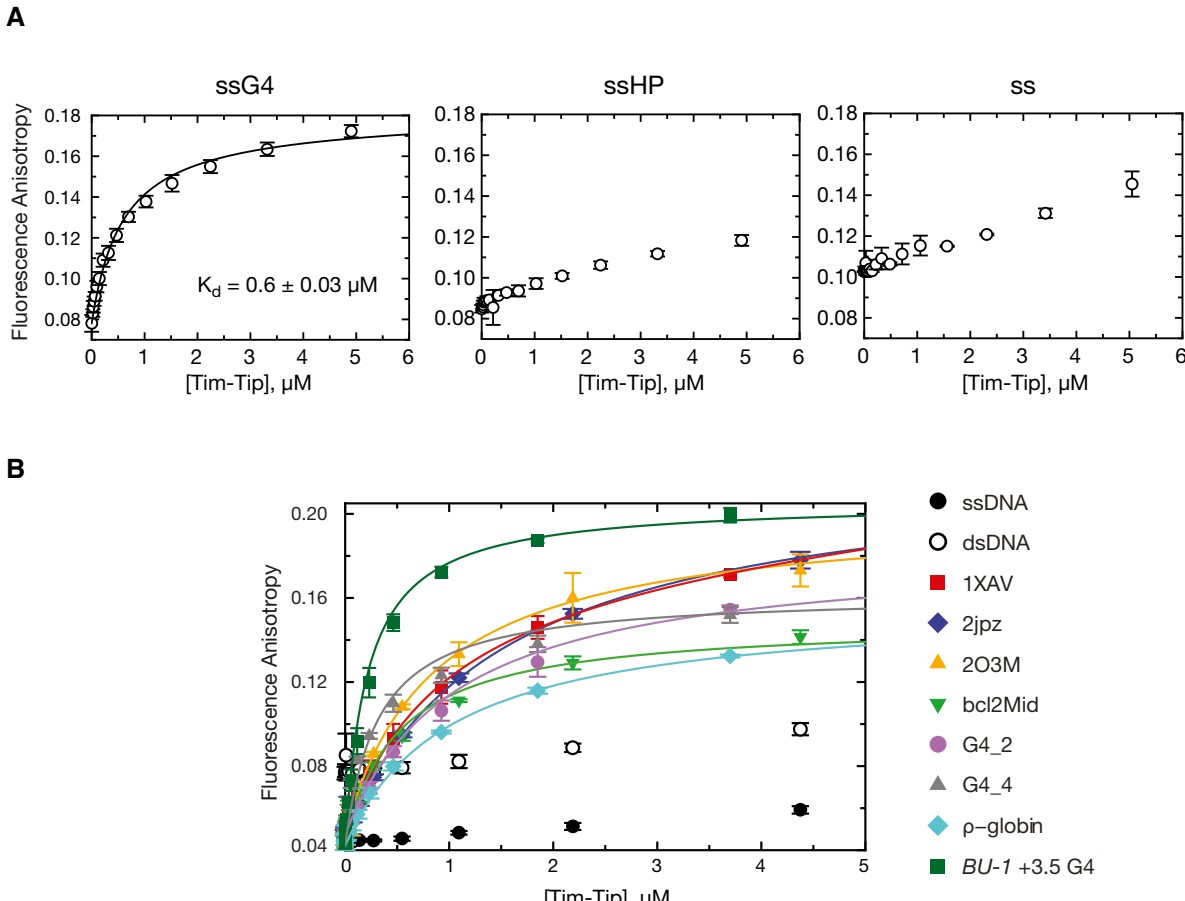

**Figure 3. The Timeless–Tipin complex shows a preference for binding G4 DNA.**

Fluorescence anisotropy was used to measure the binding affinity of Timeless–Tipin for the indicated DNA sequences.

A  ssG4: G4 flanked by single-stranded DNA; ssHP: hairpin flanked by single-stranded DNA; ss: single-stranded DNA (Appendix Table S2 for sequence details).

B  Binding affinity of Timeless–Tipin for a range of G4 DNA sequences (see Appendix Table S2 for sequence details and references). Single-stranded (ss20: 5′-6FAM-ATAAGAGTGGTTAGAGTGTA) and double-stranded (ds20: ss20 annealed to complementary sequence) DNA were also tested as controls.

Data information: Each data point is the mean of at least 3 independent experiments and the error bars indicate one SD.

FLAG-DDX11. Treatment with PDS did indeed result in more DDX11 being pulled down with PCNA (Fig 5A and Appendix Fig S4) suggesting DDX11 is recruited to chromatin after G4 stabilisation.

We next examined *BU-1* stability both in a *ddx11* mutant generated by disrupting exons 7–12 by conventional gene targeting (Abe *et al*, 2016) and in deletion mutants generated by CRISPR/Cas9 targeting to exon 4 (Appendix Fig S1). The newly generated CRISPR/Cas9-generated *ddx11* mutants are sensitive to cisplatin (Fig EV1), similar to the previously reported mutant generated by gene targeting (Abe *et al*, 2016). All *ddx11* mutants examined exhibited elevated rates of Bu-1 loss variant generation (Fig 5B). This expression instability was suppressed by complementation with full-length chicken DDX11, but not with the K87A helicase-dead form of DDX11 (the chicken equivalent of the human K50A mutation in the Walker A motif), demonstrating that the helicase activity of DDX11 is essential for G4 replication *in vivo* (Fig 5C). Importantly, the *BU-1* expression instability of *ddx11* cells is also dependent on the presence of the +3.5 G4 in *BU-1* (Fig 5C).

**DDX11 works with Timeless to prevent G4-dependent instability of BU-1 expression**

We next asked whether DDX11 and Timeless participate in the same pathway for G4 resolution at the replication fork. We generated *ddx11/timeless* double mutants by targeting Timeless in the CRISPR/Cas9-generated *ddx11* clone #1 cells. Fluctuation analysis showed that the double-mutant *ddx11/timeless* does not exhibit higher rates of Bu-1 loss variant formation, compared to the single *timeless* or *ddx11* mutants (Fig 5D), suggesting that these proteins act in the same pathway for G4 replication. Consistent with the genetic interaction between Timeless and DDX11 depending on their physical interaction, expression of DDX11 in which the "EYE" motif, required for its interaction with Timeless, is mutated to KAK (Cortone *et al*, 2018) failed to complement the instability of *BU-1* expression of *ddx11* cells (Fig 5D and Appendix Fig S5).

DDX11 is closely related to the FANCJ helicase and, like DDX11, FANCJ is able to unwind G4s and plays a prominent role in their

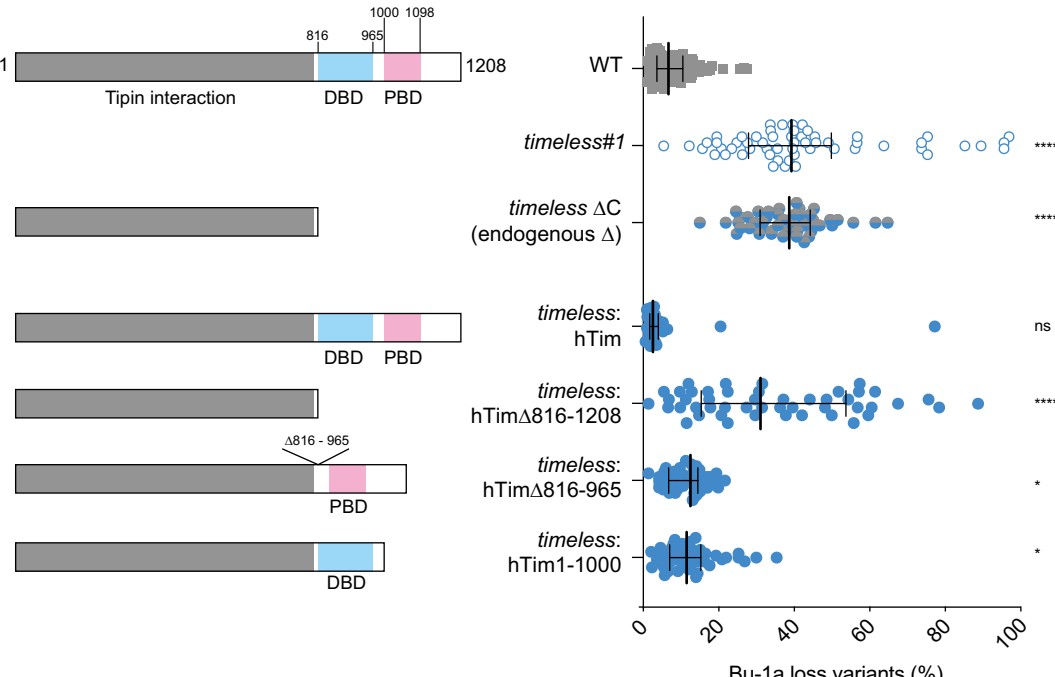

**Figure 4. The C-terminus of Timeless is required for processive G4 replication.**

Fluctuation analysis for the generation of Bu-1 loss variants. Top to bottom: wild type, *timeless* (clone 1), a *timeless* mutant (*timeless* ΔC) generated by CRISPR-Cas9 targeting exon 16 which truncates the protein removing the CTD containing both the DBD and the PARP-binding domains. Then, complementation of *timeless#1* with human Timeless (hTim), hTimΔ816–1,208 (lacking both the DBD and PBD), hTimΔ816–965 (lacking the DBD) and hTim[1:1,000], lacking the PBD. At least two independent fluctuation analyses were performed with 24–36 individual clones each cell line per repeat. Bars and whiskers represent median and interquartile range, respectively. *$P < 0.05$ and ****$P < 0.0001$; one-way ANOVA for comparison with the wild-type cells.

replication (Kruisselbrink *et al*, 2008; London *et al*, 2008; Wu *et al*, 2008; Youds *et al*, 2008). *fancj* DT40 exhibit instability of *BU-1* expression (Sarkies *et al*, 2012) that is dependent on the helicase activity of FANCJ (Fig EV4). However, a double *fancj/ddx11* mutant (Abe *et al*, 2018) exhibits significantly more *BU-1* expression instability than either single mutant (Fig 5D), suggesting that these helicases operate on the *BU-1* G4 motif independently of each other.

**Timeless and DDX11 deficiency leads to decreased proliferation and increased activation of DDR in presence of pyridostatin**

We next asked whether deficiency in Timeless and DDX11 exacerbates the globally detrimental cellular consequences of G4-binding

ligands. We treated *timeless*, *ddx11* and *ddx11/timeless* cells with the G4 ligand PDS (Rodriguez *et al*, 2008). All mutant cell lines, but not the wild type, exhibited an intrinsic reduction in doubling time with cellular proliferation further decreased in the presence of the G4 ligand (Fig 6A). We next looked at the induction of histone H2AX phosphorylation (γ-H2AX) using permeabilised cell flow cytometry. We have previously observed that PDS induces little or no γ-H2AX in wild-type DT40 cells (Guilbaud *et al*, 2017), and this was also true in our cytometric assay. However, *timeless* and *ddx11* mutants both exhibit increased levels of γ-H2AX following treatment with 4 μM PDS for 3 days compared with wild-type cells (Fig 6B and C). We confirmed our results obtained using flow cytometry by direct visualisation of the fixed

**Figure 5. DDX11 is required for processive replication in collaboration with Timeless.**

A Enhanced recruitment of DDX11 to chromatin associated PCNA following exposure to 4 μM PDS for 24 h. PCNA was precipitated from cross-linked chromatin and the immunoprecipitate blotted for FLAG-DDX11.

B Fluctuation analysis for Bu-1a loss in wild-type DT40 cells, two independent *ddx11* clones generated by CRISPR-Cas9 targeting (clones 1 and 2) and one *ddx11* clone generated by conventional homologous recombination gene targeting (clone 3). Each symbol represents the percentage of cells in an individual clone expanded for 2–3 weeks that have lost Bu-1a^high expression.

C Fluctuation analysis for Bu-1a loss variant generation in wild-type cells, *ddx11* (clone 1) cells, *ddx11* (clone 1) complemented by expression of chicken DDX11 WT cDNA, *ddx11* (clone 1) complemented by expression of helicase-dead form of chicken DDX11 (K87A) cDNA, and a *ddx11* clone generated in cells in which the endogenous +3.5 G4 has been deleted (ΔG4).

D Fluctuation analysis for Bu-1 loss in two independent *timeless/ddx11* double-mutant clones (#1 and #2), *ddx11* expressing DDX11^KAK (see Appendix Fig S3), and *fancj* and *fancj/ddx11* double mutants. Fluctuation analyses for wild type, *timeless* #1 (Fig 4) and *ddx11* #1 (Fig 5A) are shown for comparison.

Data information: In all cases, at least two independent fluctuation analyses were performed, with 24–36 individual clones each cell line per repeat. Bars and whiskers represent median and interquartile range, respectively. ****$P < 0.0001$; one-way ANOVA for comparison with wild-type cells.

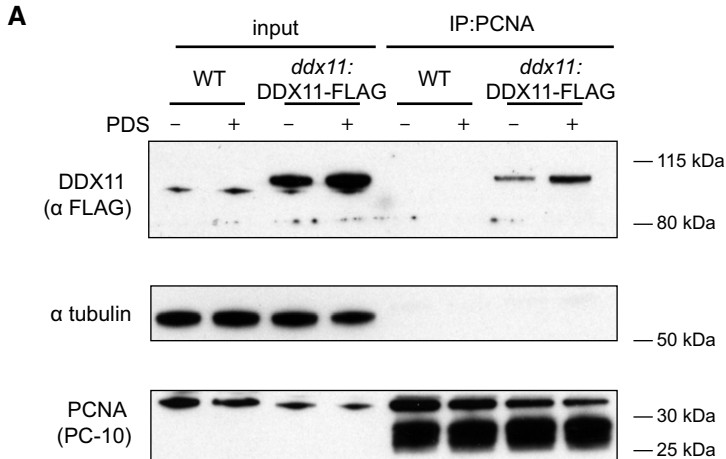

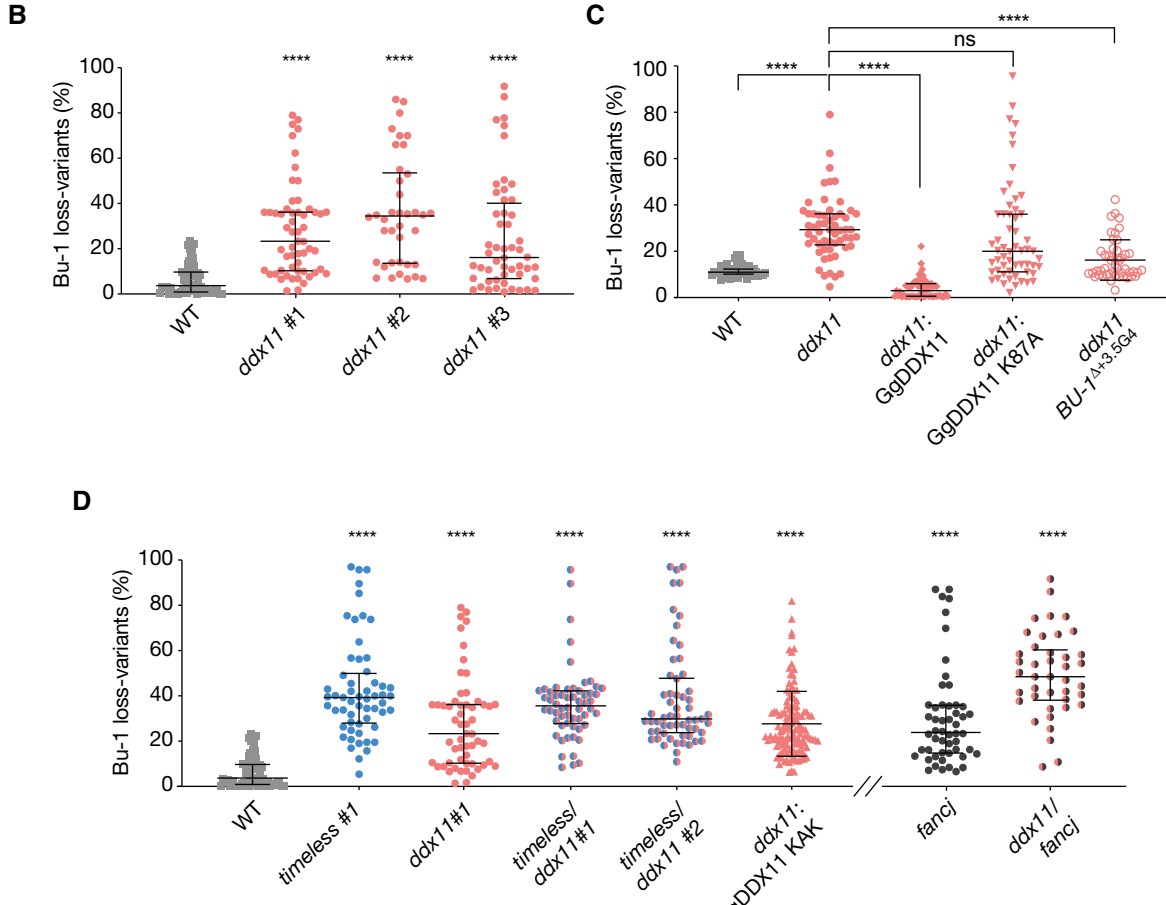

cells and quantification of γ-H2AX foci (Appendix Fig S6). γ-H2AX levels in the double *ddx11/timeless* mutant after PDS treatment were not significantly higher than the *timeless* single mutant, again supporting the observation that these two proteins operate in the same pathway for avoiding G4 ligand-induced DNA damage signalling.

## Deficiency of DDX11 and Timeless leads to dysregulation of a common set of genes harbouring G4s in close proximity to their TSS

We have previously reported that the instability of expression of the *BU-1* locus in mutants defective in G4 replication is observed in

other loci across the genome (Sarkies *et al*, 2012; Papadopoulou *et al*, 2015). We have shown that correlations between the identity of the genes dysregulated in different mutants, along with the direction and magnitude of those changes, can be used to infer genetic relationships between G4 processing enzymes (Sarkies *et al*, 2012; Papadopoulou *et al*, 2015). We applied this approach to the relationship between Timeless and DDX11, assessing gene expression changes with RNA-seq. Loss of DDX11 or Timeless induced marked changes in gene expression (*timeless* 1,752; *ddx11* 821 genes with altered expression at $P > 0.95$), with an approximately equal number of genes being up- or downregulated (Fig EV5A) in both cases. Moreover, there was a significant ($P < 2.2 \times 10^{-16}$) overlap in the identity of the deregulated genes in *timeless* and *ddx11* cells

(Fig EV5B) and a significant correlation (Spearman correlation 0.82, $P$ value $< 10^{-16}$) both in the magnitude and direction of the change in expression at the level of individual genes (Fig 7A). Consistent with our previously proposed model for replication and G4-dependent epigenetic instability (Sarkies *et al*, 2010, 2012; Schiavone *et al*, 2014), genes dysregulated in *timeless* and *ddx11* mutants are more likely to harbour a G4 close to their TSS. Using a regular expression for G4s of $(G_3N_{12})_3G_3$, we found that the region 1.5 kb around the TSS has a higher density of G4 motifs in genes dysregulated in *timeless* and *ddx11* (Fig 7B blue & red line), compared to the average for all genes (Fig 7B, black line), an observation that is not simply explained by a higher GC content (Fig EV5C). Further, a greater number of G4s are observed immediately downstream of the

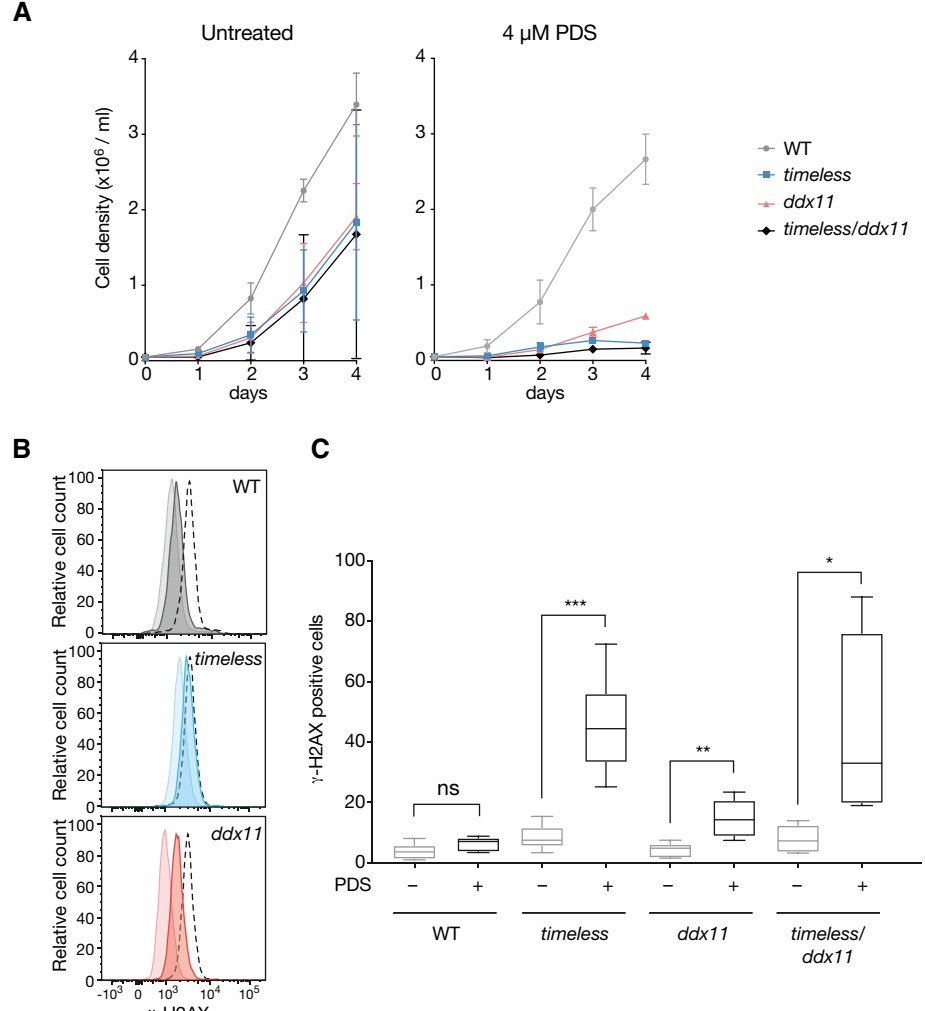

**Figure 6. Timeless and DDX11-deficient cells have impaired growth and increased H2AX phosphorylation in the presence of a G4 ligand.**

A  Growth curves for DT40 wild type, *ddx11*, *timeless* and *ddx11/timeless* cells, with and without 4 μM pyridostatin (PDS). Cells were seeded at $5 \times 10^4$ cells/ml on day 0 and the viable cells were counted each 24 h for 4 days. Bars represent SD of two independent experiments performed in duplicate. Doubling times (DMSO): WT 13 h, *timeless* 18 h, *ddx11* 16 h, *ddx11/timeless* 24 h. Doubling times (PDS): WT 13.6 h, *timeless* 27 h, *ddx11* 25.7 h, *ddx11/timeless* 47.5 h.

B  DDR signalling detected by phosphorylation of histone H2AX (γ-H2AX) by flow cytometry in untreated cells or cells exposed to 4 μM PDS for 3 days. Pale histogram, untreated; dark histogram, treated; black dotted line, positive control cells treated with 0.1 μM cisplatin, also for 3 days.

C  Quantification of γ-H2AX in DT40 wild type, *ddx11*, *timeless* and *ddx11/timeless* cells treated with 4 μM PDS for 3 days. The central band represents the median, the box the 25th–75th centile and whiskers the minimum to maximum range of three independent experiments performed in duplicate. *$P < 0.05$, **$P < 0.01$, ***$P < 0.001$ unpaired, two-tailed *t*-test for each pairwise comparison ± PDS. See also Appendix Fig S4 for immunofluorescence images of the γ-H2AX signal.

TSS in the genes dysregulated in *timeless*, *ddx11* and *fancj* mutants. These excess G4s are predominantly located on the coding strand (Fig 7C) orientated similarly to the +3.5 G4 in *BU-1* to act as a leading-strand impediment for a fork entering the locus from the 3′ end (Sarkies *et al*, 2010; Schiavone *et al*, 2014; Papadopoulou *et al*, 2015). Further, the changes in gene expression in *timeless* and *ddx11* also correlated strongly with those observed in *fancj* cells, which we have previously shown to exhibit G4-dependent epigenetic instability (Sarkies *et al*, 2012; Papadopoulou *et al*, 2015) (Fig EV5D, Spearman correlation ~ 0.9, *P* value < $10^{-16}$). These

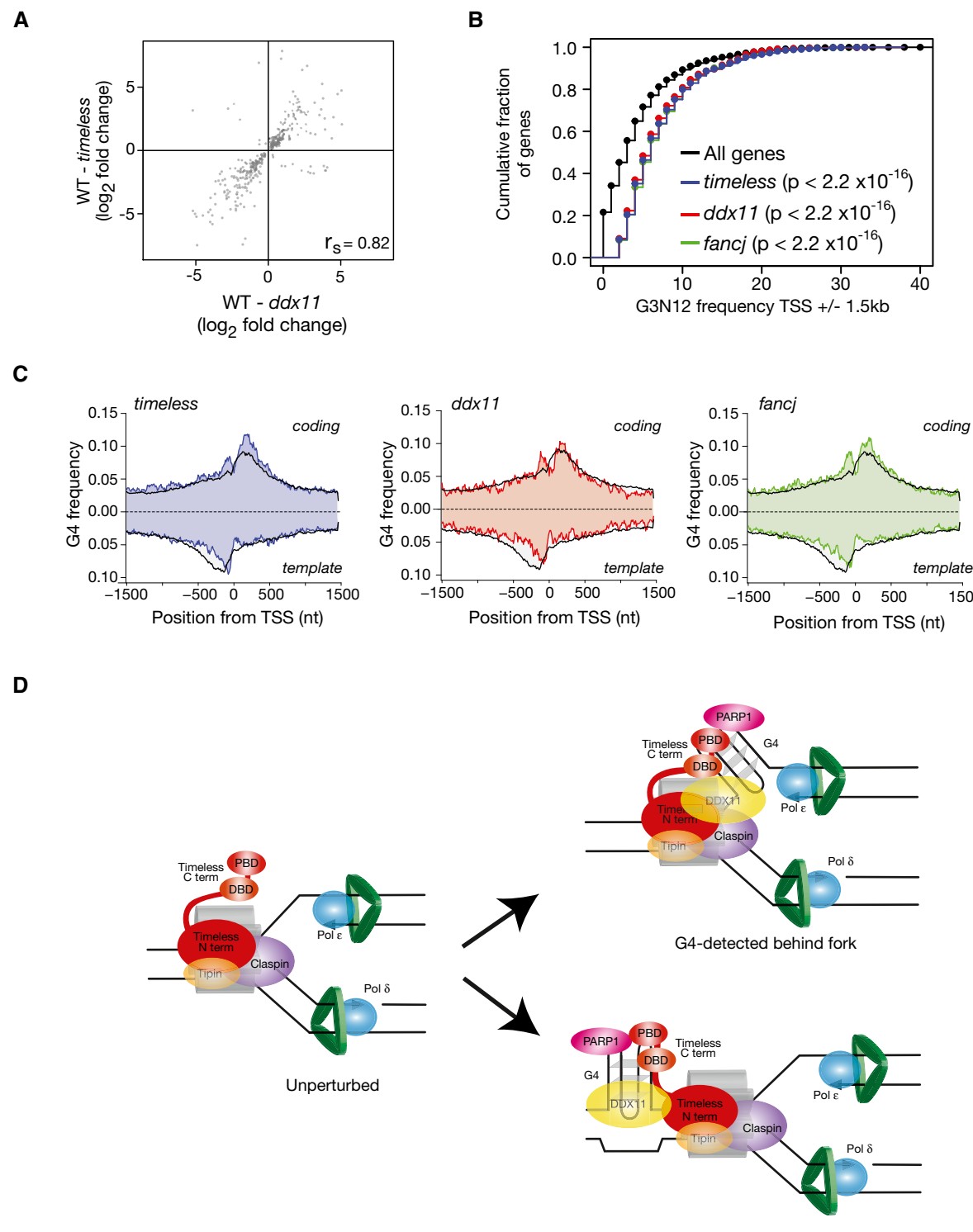

**Figure 7.**

**Figure 7.  Loss of DDX11 and Timeless leads to genome wide expression dysregulation of genes with G4s around the transcription start site (TSS).**

A   Correlation of magnitude and direction of change of genes dysregulated (relative to wild type) in *timeless vs. ddx11* DT40 cells. $r_s$ (Spearman rho) is shown for each correlation.

B   Genes dysregulated in *timeless*, *ddx11* and *fancj* mutants have a higher density of G4s around their TSS. Cumulative fraction of the genes dysregulated in *timeless* (red), *ddx11* (blue) and *fancj* (green) containing *n* (*x*-axis) G4 motifs within 1.5 kb of the TSS compared with all genes (black). *P* values calculated with the Kolmogorov–Smirnov test.

C   Metagene analysis showing G4 frequency around the TSS of genes dysregulated (up or down) in *timeless* (left panel), *ddx11* (centre panel) and *fancj* (right panel) compared with all genes (black line). G4 frequency is calculated separately for coding (above the *x*-axis) and template strands (below the *x*-axis).

D   A model for recognition and processing of replisome associated G4s by Timeless/DDX11. Current evidence suggests that Timeless is a constitutive component of the replication fork. We suggest that the C-terminus of Timeless may help detect G4s in the vicinity of the replisome by a combination of direct recognition through the DNA-binding domain and indirectly through the PARP-binding domain. It is not currently possible to distinguish whether this mechanism would operate ahead or behind the fork itself (Lerner & Sale, 2019), although recent structural evidence placing Tof1, the yeast homologue of Timeless, ahead of the fork (Baretić *et al*, 2020) would appear to make the first possibility more probable. However, for failure to resolve G4s ahead of the fork to result in uncoupling would require the CMG helicase to traverse the structure, as has been suggested for interstrand crosslinks (Sparks *et al*, 2019).

observations provide further support for a common, G4-dependent mechanism for the gene dysregulation in these three mutants. Overall, our results suggest that DDX11 and Timeless collaborate to ensure expression stability during DNA replication for a subset of genes with G4s close to their transcription start site.

## Discussion

Accumulating genetic evidence suggests that DNA secondary structure formation is a frequent challenge to vertebrate DNA replication (Lerner & Sale, 2019). Numerous factors collaborate to ensure that these structures do not result in persistent stalling of DNA synthesis or that the resulting tracts of single-stranded DNA formation are limited, for instance by efficient repriming (Schiavone *et al*, 2016). By limiting uncoupling of DNA unwinding and DNA synthesis, these mechanisms prevent both genetic and epigenetic instability.

A key to ensuring that polymerase pausing at secondary structures does not lead to extensive single-stranded DNA formation should be efficient sensing of structure formation by the replisome coupled to the recruitment of factors that will remove the structure and, if needed, reprime DNA synthesis. However, it remains to be established how secondary structure formation on the DNA template is detected by the replisome.

Timeless, along with other components of the FPC, appears to be a constitutive component of the core replisome (Gambus *et al*, 2006). Although not essential for DNA replication, the FPC increases the speed and efficiency of fork progression and DNA synthesis (Yeeles *et al*, 2017). The intimate association of the FPC with the replisome potentially puts it in an ideal place to coordinate the response of the replisome to secondary structure formation. Indeed, loss of components of the FPC leads to instability of structure-forming repeat sequences (Voineagu *et al*, 2008, 2009; Leman *et al*, 2012; Liu *et al*, 2012a; Gellon *et al*, 2019).

Here, we have shown that the C-terminus of Timeless contains a DBD that is able to bind G4 DNA more avidly than unstructured single- and double-stranded DNA *in vitro*. Using a sensitive *in vivo* assay that monitors episodes of replication pausing at a defined secondary structure-forming sequence, we show that Timeless is required to maintain processive replication of G4 DNA. This requires the C-terminus of the protein, which contains both the newly identified DBD and an adjacent domain previously shown to bind PARP (Xie *et al*, 2015; Young *et al*, 2015). The nature of the

complex formed between Timeless, Tipin and PARP1 in this context is unclear and requires further study. Timeless/Tipin and Timeless/PARP1 have been reported to form mutually exclusive complexes (Young *et al*, 2015), yet the interaction between PARP1 and Timeless is required for recruitment of both Timeless and Tipin to sites of DNA damage (Xie *et al*, 2015). Interestingly, cells expressing Timeless lacking either the DNA-binding or PARP-binding domain exhibit significant, although not complete, reversal of *BU-1* instability suggesting that there may be functional redundancy between these two domains. The precise role of PARP1 in G4 sensing during replication also remains to be explored further, but it is noteworthy that PARP1 has been previously reported to bind G4 DNA *in vitro* (Soldatenkov *et al*, 2008) suggesting, potentially, that Timeless could recognise G4s both directly through its DBD and indirectly via recruitment of PARP1 to the G4.

Timeless interacts with the G4 helicase DDX11, and this interaction stimulates the helicase activity of DDX11 (Calì *et al*, 2016). We have shown here that loss of Timeless and DDX11 are epistatic for G4-induced *BU-1* expression instability and the previously identified interaction between the two proteins (Cortone *et al*, 2018) is required, consistent with them operating as a unit, possibly with the recruitment of DDX11 to the replisome stimulated by G4 detection by Timeless. Determining the precise conformation of Timeless and DDX11 at a G4-stalled replisome remains to be determined and will represent a significant experimental challenge. Interestingly, loss of both DDX11 and the related G4-unwinding Fe–S helicase, FANCJ, results in more rapid generation of Bu-1 loss variants than either single mutant suggesting that these two helicases independently contribute to replication of the +3.5 *BU-1* G4. Our analysis of genome wide gene expression dysregulation suggests that this division of labour between DDX11/Timeless and FANCJ is likely to be extended to other G4s in the genome.

Much work remains to be done to dissect the interactions between protein factors that process G4s and other fork-stalling DNA structures, to ensure smooth progress of DNA synthesis. Mechanistically, advances in replisome structural biology should help inform where G4 sensing by Timeless takes place relative to the CMG helicase. It is reasonable to anticipate that it will detect G4s forming transiently in the negatively supercoiled DNA emerging from CMG (Fig 7D). The large size and multidomain structure of Timeless indicates that its DBD could also reach and interact with unwanted secondary structures in the DNA within a wide radius around the fork. This could potentially include those structures present in the DNA before it reaches CMG (Fig 7D). Such structures

might then be "traversed" by the helicase, analogous to its proposed behaviour at a DNA-protein crosslink (Sparks et al, 2019), before stalling the leading-strand polymerase (Lerner & Sale, 2019). In either case, detection and resolution of the structure will ensure the smooth execution of DNA synthesis, avoiding the generation of potentially deleterious ssDNA.

# Materials and Methods

### Molecular cloning

Oligonucleotide sequences can be found in Appendix Table S1. The Timeless DBD deletion (ΔDBD), PARP-binding domain truncation and ΔCTD mutant plasmids were constructed by PCR. pcDNA4-Flag-hTimeless was linearised with PCR using outward primers to allow the desired section of the cDNA to be removed. Ten fmol mutant plasmid was recircularised (through the overlapping sequences) via Gibson assembly (1 h, 50°C). Recombinant DNA was treated with 10U of DpnI to minimise carryover of the original plasmid and transformed into DH5α competent bacteria. Correct mutagenesis was confirmed by restriction digest and Sanger sequencing.

FANCJ point mutations were generated by site-directed mutagenesis using a Quick Change II XL Site-Directed Mutagenesis Kit (Agilent), according to the manufacturer's instructions, in the pEAK8-CFP-hFANCJ plasmid (generated in this study). This plasmid was generated by digesting a CFP-containing plasmid, a hWT FANCJ cDNA-plasmid and a pEAK8 backbone plasmid with HindIII and SalI, SalI and NotI and HindIII and NotI, respectively. All fragments were gel-purified with a QIAEX II Gel Extraction Kit (QIAGEN) and ligated in a three-way ligation reaction using a Rapid Ligation Kit (Roche) for 30 min at room temperature.

### Cell lines and transfections

All DT40 cell lines were cultured at 37°C in a humidified 10% $CO_2$ atmosphere in RPMI 1640 with Glutamax (Thermo Fisher Scientific) supplemented with 7% foetal bovine serum (Thermo Fisher Scientific), 3% chicken serum (Sigma-Aldrich), 50 μM 2-mercaptoethanol and 1% penicillin/streptomycin, as previously described (Simpson & Sale, 2003). HEK293T cells were cultured at 37°C in a humidified 5% $CO_2$ atmosphere in DMEM (Thermo Fisher Scientific) supplemented with 10% foetal bovine serum and penicillin/streptomycin.

DT40 ddx11 (CRISPR), timeless and ddx11/timeless mutants were generated in this study by CRISPR-Cas9-mediated gene disruption. DT40 WT $BU-1^{\Delta G4}$ cells were generated by deleting the +3.5 G4 motif from both alleles of the BU-1 locus (Schiavone et al, 2014). Other mutant lines have also been described previously: fancj (Sarkies et al, 2012), ddx11, fancj/ddx11 and tipin (Abe et al, 2016, 2018).

Complemented ddx11 cells were obtained by transfecting the KO cell lines with pEGFP-C1 (Clontech) harbouring chicken DDX11 cDNA or a helicase-dead variant (K87A) (Abe et al, 2016) with selection of G418-resistant (2 mg/ml) clones followed by screening for GFP expression by flow cytometry. The EYE motif in DDX11 (Cortone et al, 2018) was mutated to KAK by site-directed mutagenesis using the Quick Change II XL Site-Directed Mutagenesis Kit

(Agilent) in pcDNA3-Flag-hDDX11 plasmid (Cortone et al, 2018), a kind gift from Francesca Pisani. Complementation of timeless was achieved by transfection of plasmids encoding the human Timeless WT cDNA (Addgene plasmid 22887), or truncated versions thereof, selected with zeocin (1 mg/ml) followed by screening for Flag-positive clones by Western blot (WB). Complemented fancj cells were obtained as follows. First, $fancj^{+/-}$ cells harbouring a tamoxifen-regulatable Cre recombinase, Mer-Cre-Mer (Zhang et al, 1996) were transfected with pXSPN (Ross et al, 2005) with wild-type human YFP-FANCJ flanked by loxP sites. The second allele of the FANCJ locus was then disrupted with a previously described targeting construct (Sarkies et al, 2012). For testing the effect of FANCJ mutations, a second pXPSN plasmid was introduced harbouring CFP-FANCJ[mut] without loxP sites, where "mut" is the desired mutation. Treatment with tamoxifen results in excision of the wild-type FANCJ transgene leaving the cells either FANCJ-deficient or solely expressing FANCJ[mut].

Transfections for CRISPR-Cas9 targeting were performed using a Neon electroporator (Thermo Fisher Scientific), $20 \times 10^6$ cells and 20 μg plasmid DNA and three pulses of 1,400 V with intervals of 10 msec. For cDNA expression, either the Neon, with the same conditions, or a BioRad electroporator with 1 pulse of 250 V 950 μF 100 Ω in 4 mm cuvettes was used.

### Gene disruption using CRISPR-Cas9

Guide RNA sequences used for disrupting DDX11 and TIMELESS in DT40 cells are listed in Appendix Table S1. Guide RNAs were designed using the CRISPOR online tool of the Zhang lab (https://zlab.bio/guide-design-resources). Each guide was cloned into the pSpCas9(BB)-2A-GFP (PX458) plasmid (Ran et al, 2013) and transfected as described above. Twenty-four hours after transfection, cells were collected by centrifugation at 400 g for 5 min and sorted at single live (propidium iodide negative), GFP-positive cell per well in 96-well plates using a MoFlo (Beckman Coulter) or a Synergy (Sony) cell sorter. Cells were grown for 2 weeks, and clones were collected and genotyped by Sanger sequencing of the targeted region amplified by PCR, gel-purified with a QIAquick Gel Extraction Kit (QIAGEN) and blunt-cloned using Zero Blunt TOPO PCR Cloning Kit (Thermo Fisher Scientific). Sequences were aligned against the WT DT40 genome using MacVector software.

### Bu-1 staining and fluctuation analysis for generation of Bu-1 loss variants

Bu-1 staining and fluctuation analysis were performed as described previously (Schiavone et al, 2014; Guilbaud et al, 2017). Briefly, cells (confluency between $0.4–2 \times 10^6$) were directly stained with anti-Bu-1 conjugated with phycoerythrin (Santa Cruz 5K98-PE 70447 or Invitrogen 21-1A4-PE MA5-28754) at 1:100 dilution for 10 min at room temperature. Cells were analysed by flow cytometry using an LSRII flow cytometer (BD Biosciences) and the FlowJo software. Bu-1 expression in single live cells was gated using side scatter (SSC) in the y-axis and PE fluorescence in the x-axis. To perform fluctuation analysis, Bu-1-positive single cells were sorted and grown for ~ 20 generations before staining and flow cytometry analysis as described above.

    

## G-quadruplex ligand

The small molecule pyridostatin (PDS) (Rodriguez *et al*, 2008) was purchased from Sigma-Aldrich.

## Cell proliferation and viability assays

$5 \times 10^5$ cells were seeded in 10 cm dishes containing 10 ml of media with G4 ligand/DMSO. Living cells were counted at 24, 48, 72 and 96 h on a Vi-Cell cell counter (Beckman Coulter). Doubling time was calculated during the exponential growth phase using the formula Doubling Time = duration *log (2)/(log (Final Concentration) − log (Initial Concentration)).

Cell viability was determined 72 h after PDS or cisplatin treatment using a CellTiter 96 AQueous One Solution Cell Proliferation Assay (Promega), according to the manufacturer's instructions. $1 \times 10^5$ cells/well were seeded in 96-well plates in 200 μl media containing PDS or cisplatin. Seventy-two h later, 20 μl of MTS reagent was added to 100 μl cells and incubated for 2–4 h at 37°C. Absorbance was measured at 492 nm (formazan salt) in a PHER-Astar plate reader (BMG Labtech). The percentage of cell viability was calculated as follows: Cell viability (%) = (Test absorbance/ Control absorbance)*100.

## Flow cytometry and microscopy to monitor γ-H2AX

Cells were collected, fixed with 1% paraformaldehyde for 15 min in ice, washed with PBS and fixed with 70% ethanol for at least 24 h at −20°C. Cells were blocked and permeabilised in BD wash/permeabilisation buffer (BD Biosciences) and then incubated with anti-γ-H2AX antibody (05-636, Merck, diluted at 1:500 in BD Buffer) for 2 h at room temperature. Samples were washed twice with BD Buffer, incubated with anti-mouse AF594 antibody (A11062, Thermo Fisher Scientific, diluted at 1:200 in BD Buffer) for 1 h at room temperature in the dark and then washed twice with BD Buffer. Cells were resuspended in PBS containing 0.5% BSA and 1 μg/ml DAPI. Cells were analysed by flow cytometry using an LSRII flow cytometer (Beckman Coulter) and the FlowJo software. A gate for γ-H2AX-positive cells was defined based on untreated cells that were considered negative for γ-H2AX staining (see Fig 6B). Cells for microscopy were prepared in exactly the same way as for cytometry with a drop of the final suspension of fixed and stained cells being directly visualised using a Nikon C1si equipped with a Hamamatsu ORCA-Flash4.0 LT$^+$ cooled CMOS camera controlled by Nikon NIS Elements Software. γH2AX foci were quantified using Fiji (Schindelin *et al*, 2012). Briefly, the blue channel (DAPI staining) was used to define masks to outline cell nuclei with the "Threshold" function. Merged nuclei were split with the "Watershed" function. γH2AX foci were identified using the "Find Maxima" function on the green channel (γH2AX staining), and nuclear foci were counted by applying the previously defined masks. A minimum of 50 cells per conditions was considered for statistical analysis. Differences between conditions were tested using Student's *t*-tests.

## Protein expression and purification

Full-length human Timeless (1–1,208) was amplified from I.M.A.G.E cDNA clone IRATp970C0576D (Source Bioscience) and cloned into pRSF-Duet1 (Novagen) with an N-terminal His$_{14}$-SUMO tag. Our previous structural analysis had demonstrated that residues 239–330 comprise a large disordered loop within the folded N-terminal region of Timeless (Holzer *et al*, 2017) and their removal improved dramatically the biochemical behaviour of the protein; they were therefore replaced by a short linker consisting of residues (Gly–Ser–Thr)$_2$. All experiments reported here were performed with this optimised Timeless construct. The full-length human Tipin gene (1–301) was codon-optimised for *Escherichia coli* expression (Life Technologies) and cloned into the pGAT3 vector for expression fused to an N-terminal His$_6$-GST tag (Peränen *et al*, 1996). Timeless and Tipin were co-expressed in *E. coli* Rosetta2 (DE3). The complex was purified using Ni-NTA agarose (Qiagen) followed by cleavage of the tags with TEV and SUMO proteases, ion exchange chromatography (HiTrap Q HP, GE Healthcare) and size-exclusion chromatography (Superdex 200 16/60, GE Healthcare). The DBD of human Timeless (residues 816–954) was cloned into pRSF-Duet1 (Novagen) with an N-terminal His$_{14}$-SUMO tag and expressed in *E. coli* Rosetta2 (DE3). Purification entailed Ni-NTA agarose (Qiagen), His$_{14}$-SUMO tag cleavage with SUMO protease, Ni-NTA recapture of the cleaved tag and size-exclusion chromatography (Superdex 75 16/60, GE Healthcare). Timeless 883–947 (DBD C-term) was expressed and purified in the same way as DBD.

## X-ray crystal structure determination

The DBD C-term was crystallised by vapour diffusion, mixing equal volumes of protein at 800 μM and a solution of 200 mM sodium formate and 18% PEG 3350 at 19°C. For cryoprotection of the crystals, the mother liquor was replaced with a 2:2:1 (volume ratio) mixture of reservoir solution, protein buffer and 100% glycerol. X-ray diffraction data for a native crystal were collected at the I24 beamline at Diamond Light Source, Oxford, UK. The dataset was processed with XDS (Kabsch, 2010) in space group P6$_5$ to a resolution of 1.15 Å.

For structure determination by anomalous scattering, crystals were grown of Se-Met labelled DBD. Selenomethionine was incorporated in the DBD using metabolic inhibition in the presence of Se-Met during bacterial expression in minimal media. Diffraction data at the peak wavelength of 0.9793 Å were collected at the Proxima2A beamline of the Soleil Synchrotron, Gif-sur-Yvette, France. The data were processed to 2.4 Å with XDS (Kabsch, 2010) in the same space group and cell dimensions as for the native crystal. The structure was solved exploiting the anomalous signal of the peak dataset using Autosol (Phenix) (Zwart *et al*, 2008). A largely complete Se-Met model was used to solve the native dataset by molecular replacement using PHASER (McCoy *et al*, 2007). The model of the resulting solution was subsequently extended by Autobuild (Phenix) (Zwart *et al*, 2008) and completed by iterative cycles of manual building and model refinement in Coot (Emsley & Cowtan, 2004) and Phenix (Zwart *et al*, 2008). The structural figures were generated with Chimera (Pettersen *et al*, 2004). Data collection and refinement statistics are given in Appendix Table S3.

## NMR spectroscopy

For expression of $^{15}$N- or $^{13}$C, $^{15}$N-labelled Timeless 816–954, $^{13}$C$_6$-glucose and/or $^{15}$NH$_4$Cl was used as the sole carbon/nitrogen

source in M9 minimal medium. NMR measurements were made on ~ 0.5 mM solutions in 10% $^2$H$_2$O, 10 mM phosphate (pH 6.4), 40 mM KCl and 0.5 mM EDTA. Experiments were recorded at 25°C on Bruker AVANCE III 600 or 800 MHz spectrometers equipped with QCI or TXI cryoprobes. Data were processed using the AZARA suite of programs (v. 2.8, © 1993–2019; Wayne Boucher and Department of Biochemistry, University of Cambridge, unpublished). Backbone assignments were derived from established versions (Cavanagh *et al*, 2006) of HNCA, HN(CO)CA, HN(CO)CACB, HNCACB and HNCO experiments acquired with non-uniform sampling, and side chains using 3D (H)CC(CO)NH-TOCSY, TOCSY-$^{15}$N-HSQC, NOESY-$^{15}$N-HSQC, HC(C)H-TOCSY and NOESY-$^{13}$C-HSQC. Assignment was carried out using CcpNmr Analysis v. 2.4 (Vranken *et al*, 2005). For the heteronuclear NOE experiments, either 4 s of $^1$H saturation using a 120° pulse train or a 4-s delay was employed prior to the first $^{15}$N pulse. Inter-proton distance restraints were derived from NOE peak heights in the 3D NOESY spectra using the relaxation matrix method (which accounts for spin diffusion) and $r^{-6}$ summation for ambiguous NOEs. Backbone $\phi$ and $\psi$ torsion-angle restraints were obtained using TALOS$^+$ (Shen *et al*, 2009). The restraints provided the input for the iterative assignment protocol ARIA v.1.2 (Linge *et al*, 2003). For the final iteration, 100 structures were calculated with the violation tolerance set to 0.1 Å. After the last iteration, the 20 lowest energy structures were subjected to a final water refinement to give an ensemble of 20 structures. Torsion-angle molecular dynamics simulations were performed using CNS (Brünger *et al*, 1998). The simulated annealing protocol used to calculate each structure included 60,000 molecular dynamics steps, including those for refinement, and two cooling stages (to 1,000 K and 50 K). A summary of the restraints used in the calculation of the structure and characterisation of the ensemble of energy-minimised structures is presented in Appendix Table S4.

### EMSA-based DNA-binding experiments

G-quadruplex sequences (Appendix Table S2) were folded by heating the DNA to 95°C followed by gradual cooling to 10°C, in TE buffer supplemented with 150 mM KCl. Reactions contained 5 μM Timeless–Tipin and 5 μM 6FAM-labelled DNA in 25 mM HEPES pH 7.1, 150 mM KCl, 5 mM MgCl$_2$ and 1 mM DTT. Reactions were incubated on ice for 30 min, after which native loading dye was added and the samples analysed on a 1% agarose gel in EMSA buffer (0.5× TBE, 10 mM KCl). Gels were run for 1 h at 50 V and 4°C, and bands were visualised under UV light.

### Fluorescence anisotropy analysis of DBD domain and Timeless–Tipin complex binding to DNA

DNA sequences were annealed by heating to 95°C followed by gradual cooling to 10°C, using a thermocycler, in TE buffer supplemented with 150 mM KCl. Fluorescence anisotropy measurements were recorded at 25°C in a plate reader (PHERAstar FS; BMG Labtech). Excitation for Cy3-3′-labelled DNA was at 540 nm and emission at 590 nm (20 nm bandwidth for both). Excitation for FAM6-5′-labelled DNA was at 485 nm and emission at 520 nm (10 nm bandwidth for both). For DBD titrations, each well contained 10 nM 3′Cy3-labelled DNA and increasing concentrations

of DBD protein in assay buffer (25 mM HEPES-NaOH pH 7.2, 150 mM KCl, 0.5 mM EDTA, 0.005% Tween-20). For Timeless–Tipin complex titrations, each well contained 10 nM 6FAM-labelled DNA and increasing concentrations of the complex in assay buffer (25 mM HEPES pH 7.1, 150 mM KCl, 1 mM DTT, 5 mM MgCl$_2$). The voltage gains and focal heights were adjusted using the instrument software using free fluorescein as reference (35 mP), and data collection was set to 200 flashes. Each data point is the mean of at least three independent measurements. The averaged data were analysed using non-linear fits of ligand-depletion binding isotherms adapted for fluorescence anisotropy measurements, assuming one to one binding model and using the *ProFit* software package (Quantum Soft). No corrections were necessary for changes in the quantum yield as a function of protein concentration.

### Co-immunoprecipitation and WB

For whole-cell extracts, the protein samples were lysed in lysis buffer (50 mM Tris–HCl pH 7.5, 20 mM NaCl, 1 mM MgCl$_2$, 0.1% SDS, protease inhibitors (Complete Protease Inhibitor Cocktail Tablets, Merck), phosphatase inhibitors (Halt Phosphatase Inhibitor Cocktail, Thermo Fisher Scientific) and 250 U/ml benzonase (Merck)) for 15 min at room temperature under agitation and cleared by centrifugation at 16,000 *g* for 20 min at 4°C. Samples were quantified using the Bradford method, and approximately 30 μg were mixed with 5× SDS–Page Sample Buffer (Jena Bioscience), boiled for 3 min, fractionated in 4–12 or 10% Nu-PAGE Bis-Tris gels (Thermo Fisher Scientific) in MOPS (200 mM MOP, 50 mM Sodium Acetate, 10 mM Na$_2$EDTA) and transferred to nitrocellulose membranes using an iBlot2 apparatus (Thermo Fisher Scientific) at 25 V for 7 min. Membranes were blocked for 1 h at room temperature with TBS 0.1% Tween-20 5% skimmed milk. Membranes were incubated with primary antibodies diluted in TBS-T 5% milk overnight at 4°C under agitation, washed with three washes of 10 min each with TBS-T and incubated with HRP-conjugated secondary antibodies at room temperature for 1 h, followed by three washes with TBS-T. Membranes were developed with Immobilon Crescendo Western HRP substrate (Merck) for 1 min and exposed to X-ray films.

For the Co-IP experiments with Timeless, pcDNA4-Flag-Timeless plasmids (WT and truncated versions) were co-transfected into 2 × 10$^6$ HEK293T cells using Lipofectamine 2000 (Thermo Fisher Scientific) and seeded in 10 cm dishes 16 h prior to transfection (80% confluency at transfection) with pcDNA3-hDDX11 (Abe *et al*, 2016) (30 μg total DNA). Twenty-four hours after transfection, cells were detached with trypsin and washed in cold PBS. Pellets were lysed in NETN-M buffer with glycerol (150 mM NaCl, 50 mM Tris–HCl pH 8, 0.5% Igepal, 1 mM EDTA, 5 mM MgCl$_2$ and 10% glycerol, supplemented with protease and phosphatase inhibitors) and benzonase for 15 min at room temperature under agitation. Samples were sonicated with eight cycles of 30 s with 30-s intervals at the low setting in a Bioruptor Plus water bath sonicator (Diagenode) and centrifuged for 15 min at 16,000 *g* at 4°C. Supernatant was collected and quantified, and 2 mg of total cell extract was mixed with 30 μl Flag-M2 magnetic beads (M8823, Sigma-Aldrich). Samples were incubated at 4°C for 2 h under agitation. Beads were then washed five times for 5 min each with lysis buffer, and proteins were eluted by boiling with Sample Buffer Laemmli 2×

(Sigma-Aldrich) for 5 min at 95°C. Samples were subjected to analysis by WB as described above, using the indicated antibodies.

For ChIP of DDX11 with PCNA, $24–36 \times 10^6$ DT40 cells were cross-linked with 1% formaldehyde for 5 min at room temperature, quenched with 0.2 M glycine and then lysed with hypotonic lysis buffer (10 mM Tris–HCl pH 7.4, 10 mM NaCl, 3 mM $MgCl_2$, 0.5% NP40) at room temperature. Crude chromatin extract was resuspended in sonication buffer (50 mM Tris–HCl pH 8.0, 1% SDS, 10 mM EDTA), solubilised by sonication in Bioruptor plus (30 cycles of 30″ On/Off, high setting at 4°C) and then diluted tenfold with IP dilution buffer (16.7 mM Tris–HCl pH 7.4, 167 mM NaCl, 1.2 mM EDTA, 1.1% Triton X-100). PCNA-bound chromatin was immunoprecipitated with 0.5 μg PC-10 antibody (Santa Cruz Biotechnology, sc-56) overnight. Immunocomplexes were captured with 40 μl of Protein G magnetic beads (CST, #9006) and washed extensively with low-salt, high-salt and LiCl wash buffers and TBS-T. Crosslinks were reverted and proteins eluted by boiling the beads for 10 min at 95°C with 1.5× Laemmli buffer.

Primary antibodies used: mouse anti-Flag (M2, F3165, Sigma), mouse anti-DDX11 (D-2 271711, Santa Cruz), mouse anti-α-tubulin (T6074, Sigma), mouse anti-human PCNA PC10 (sc-56, Santa Cruz) and mouse anti-human PCNA C-20 (sc-9857, Santa Cruz). Secondary antibody: goat anti-mouse HRP (P0447 Dako).

### RNA extraction and RNA-seq library preparation

RNA was extracted from three independent cell populations using the RNeasy Mini Kit (QIAGEN), according to the manufacturer's instructions. RNA sample quality was checked in BioAnalyzer RNA Pico chips (Agilent), and only high-quality samples (RIN > 7) were used to build the libraries. Seven hundred fifty nanogram RNA was used to build the next-generation sequencing libraries with the NEBNext Ultra II RNA Library Prep Kit for Illumina (New England BioLabs), NEBNext Poly(A) mRNA Magnetic Isolation Module and NEBNext Multiplex Oligos for Illumina (Index Primers Set 1 and 2), according to the manufacturer's instructions. Library quality was checked in BioAnalyzer DNA High Sensitivity chips (Agilent) and quantified using the KAPA Library Quanti Kit (Illumina) Universal qPCR Mix (KAPA Biosystems) on an ABI Prism ViiA 7 Real-Time PCR System (Thermo Fisher Scientific). Libraries were sequenced on a HiSeq4000 (Illumina).

### RNA-seq library alignment

RNA sequencing reads were aligned to the GRCg6a chicken genome using Bowtie 2 (Langmead & Salzberg, 2012). Expression estimates (transcripts per million; TPM) were calculated using the rsem-calculate-expression command of RSEM (Li & Dewey, 2011) and the version 96 of the chicken genome gene annotation from Ensembl. Only transcripts with an expression above or equal at 1 TPM were considered for further analysis. Gene expression matrixes were built using rsem-generate-data-matrix and three independent biological replicates. Differentially expressed genes were called using EBseq (Leng *et al*, 2013). Change in gene expression was estimated using the EBSeq posterior Fold Change (PostFC) value, and genes were considered differently expressed when the EBseq posterior probability of being differentially expressed (PPDE) was above or equal at 0.95.

### Promoter sequence analysis

Gene promoter sequences were recovered from the GRCg6a chicken genome assembly using the getfasta command of BEDTools (Quinlan & Hall, 2010). Sequence analyses were performed using custom scripts in the R environment (http://www.R-project.org/). Assessment of the enrichment of G4s within the promoter of DDX11- and TIMELESS-dependent genes was based on regular expression matching algorithms to monitor the cumulative count of the G3N12 motif within promoters. This was computed by identifying the number of sequences of the form $d(G_{3+}N_{1–12}G_{3+}N_{1–12}G_{3+}N_{1–12}G_{3+})$, where N is any base, which represents the loose definition of G4 forming sequences (Huppert & Balasubramanian, 2005). Densities of G4-forming sequences around promoters were computed by assessing the number of promoter sequences containing sequences of the form $d(G_{3+}N_1+G_{3+}N_1+G_{3+}N_1+G_{3+})$ in windows of 50 nucleotides sliding by 10 nucleotides normalised to the total number of promoter sequences analysed.

### Statistical analysis of RNA-seq data

Data were analysed and statistics performed in the R environment. Overlaps between gene lists were tested using Fisher's exact tests. Differences between distributions were tested using Kolmogorov–Smirnov tests.

## Data availability

The datasets produced in this study are available in the following databases: Coordinates and structure factors have been deposited in the Protein Data Bank (https://www.rcsb.org) under accession codes 6T9Q (http://www.rcsb.org/pdb/explore/explore.do?structureId=6T9Q) for the crystal structure of the C-terminal repeat of DBD and 6TAZ (http://www.rcsb.org/pdb/explore/explore.do?structureId=6TAZ; BMRB ID 34443) for the NMR structure of the DBD. The RNA sequencing data have been deposited in GEO (https://www.ncbi.nlm.nih.gov/geo/) with accession number GSE139256 (http://www.ncbi.nlm.nih.gov/geo/query/acc.cgi?acc=GSE139256).

**Expanded View** for this article is available online.

### Acknowledgements

The authors thank M. Daly and F. Zhang in the LMB flow cytometry facility for cell sorting, Anno Koetje for help with the DNA-binding experiments, Joe Yeeles for discussions and members of both the Sale and Pellegrini groups for critical reading of the manuscript. Work in the Sale group is supported by a core grant to the LMB by the MRC (U105178808). L.K.L. received a 1-year Science Without Borders postdoctoral funding from the Coordenação de Aperfeiçoamento de Pessoal de Nível Superior (CAPES, Brasília, DF, Brazil). Work in the Pellegrini group is supported by a Wellcome Trust investigator award to L.P. (104641/Z/14/Z), a Boehringer-Ingelheim Fonds PhD fellowship and awards from the Janggen-Poehn-Stiftung and the Swiss National Science Foundation to S.H.

### Author contributions

JES, LP, LKL and SH conceived the overall project. LKL generated the cell lines and performed the molecular biology experiments (flow cytometry, survival, IP,

RNA-seq). SH expressed and purified the Timeless-Tipin complex, the DBD and the DBD C-term domain, determined the X-ray crystal structure of the C-term domain, and performed the initial characterisation of the DNA-binding properties of Timeless-Tipin and the DBD. SS constructed the truncated versions of Timeless plasmids, performed ChIP experiments and some fluctuation analyses. DS generated the mer-cre-mer FANCJ cell line. CBE performed the γ-H2AX IF microscopy. PM performed the RNA-seq data analysis. MLK, AB and JDM performed the DNA-binding experiments. KS and SH determined the NMR structure of the DBD. DB provided key KO cell lines. LKL, JES and LP wrote the initial manuscript with assistance and editing by all authors.

## Conflict of interest

The authors declare that they have no conflict of interest.

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
