## [Review Process File · The EMBO Journal]

Timeless couples G-quadruplex detection with processing by DDX11 helicase during DNA replication

Leticia Koch Lerner, Sandro Holzer, Mairi Kilkenny, Saša Šviković, Pierre Murat, Davide Schiavone, Cara Eldridge, Alice Bittleston, Joseph Maman, Dana Brnzei, Katherine Stott, Luca Pellegrini, and Julian Sale

DOI: [10.15252/embj.2019104185](https://doi.org/10.15252/embj.2019104185)

Corresponding author(s): [Julian Sale \(jes@mrc-lmb.cam.ac.uk\)](mailto:jes@mrc-lmb.cam.ac.uk), [Luca Pellegrini \(lp212@cam.ac.uk\)](mailto:lp212@cam.ac.uk)

Review Timeline:

Submission Date:	4th Dec 19
Editorial Decision:	8th Jan 20
Revision Received:	10th Jun 20
Editorial Decision:	24th Jun 20
Revision Received:	25th Jun 20
Accepted:	26th Jun 20

Editor: *Hartmut Vodermaier*

Transaction Report:

Dr. Julian E. Sale
MRC Laboratory of Molecular Biology
Division of Protein & Nucleic Acid Chemistry
Francis Crick Avenue
Cambridge Biomedical Campus
Cambridge CB2 0QH
United Kingdom

8th Jan 2020

Re: EMBOJ-2019-104185

Timeless couples G quadruplex detection with processing by DDX11 during DNA replication

Thank you for submitting your manuscript on Timeless/DDX11 interplay during G-quadruplex motif replication for our editorial consideration. Three referees have now returned their detailed reports, copied below for your information, and have further commented on each others' reviews. I am pleased to say that in light of these assessments and further consultations, where two referees reaffirmed their positive views on the novelty and significance of the present findings, we would be interested in considering this work further for EMBO Journal publication, pending satisfactory revision of a number of specific concerns raised in the reports.

In particular, I would ask you to address the various major and minor issues raised by referees 1 and 2. It will also be important to follow up referee 3's queries on whether DDX11 forms a constitutive complex with Timeless at replication forks and whether Timeless, Tipin and PARP1 coexist in the same complex. On the other hand, mapping and mutating of Timeless interaction regions would seem redundant with the complementation assays using DDX11 "EYE" motif mutants as suggested by referee 1 (point 2). Similarly, I feel that detailed determination of G4 binding by Timeless may be beyond the scope of this first description of such recognition. Nevertheless, given our policy to allow only a single round of (major) revision, it will still be important to diligently respond to all raised points at this stage. Should you have any specific questions/comments in this regard, or would like to request an extension of the revisions period, please do not hesitate to contact me any time.

Further information on preparing and uploading a revised manuscript can be found below and in our Guide to Authors.

Thank you again for the opportunity to consider this work for The EMBO Journal! I look forward to your revision.

REFEREE REPORTS

Referee #1:

This work, carried out in collaboration by the laboratories of Julian Sale and Luca Pellegrini, describes a previously unrecognised DNA-binding domain (DBD) of Timeless, a component of the replication fork-protection complex, that shows high affinity for G-quadruplex-forming DNA oligonucleotides as revealed by *in vitro* binding assays. The Authors have carried out a biochemical and structural analysis of the Timeless DBD and have examined the role of this domain in assisting smooth replication of a DT40 genomic locus that harbours a sequence with the potential of forming a G-quadruplex structure downstream the BU-1 gene promoter. In previous studies by the Sale's laboratory, prolonged stalling of replication forks at this G-quadruplex structure was demonstrated to cause an altered epigenetic mark transmission at this genomic site and a consequently reduced expression of the surface glycoprotein encoded by the BU-1 gene, a change that can be directly monitored by a flow cytometry-based assay. Besides, the Authors shows that Timeless acts in concert with the DDX11 DNA helicase in promoting replication through this G-quadruplex-containing DNA template, in agreement with a previous report that revealed a direct physical and functional interaction between Timeless and DDX11 in human cells during DNA replication in stressful conditions [Calì et al., 2016, NAR]. At the end of the manuscript, a model is proposed where Timeless senses and binds G-quadruplex structures that arise at the DNA replication forks and recruits DDX11 to promote resolution of these structures in a pathway found to be redundant to the one of the paralogous FANCI DNA helicase.

This study is extremely novel and significant and the reported findings are of paramount interest in the field of genome stability/chromosome dynamics, especially considering that the role of the mammalian fork-protection complex in assisting progression of the replication machinery through difficult-to-replicate DNA templates is not fully understood from a mechanistic standpoint. The results of this work and the derived conclusions are persuasive. Nonetheless, before publication in EMBO Journal, some additional experiments need to be carried out to reinforce conclusions and the proposed model about the roles played by Timeless, DDX11 and PARP1 when DNA replication forks are stalled at sites forming G-quadruplex structures, as specified below.

Major points:

1. The possible role of PARP1 in the resolution of G-quadruplex DNA structures at the replication forks needs to be further investigated. The Authors have found that the Timeless PARP1-binding domain (PBD) is required for a complete rescue of the BU-1 genetic instability in avian cells. I suggest that the Authors examine if PARP1 operates in this or a redundant pathway of G-quadruplex resolution and if the PARP1 catalytic activity is required for G-quadruplex processing. This would require the use of PARP1-KO DT40 cells already described [Ooka et al., PLoS One, 2018] and complementation assays with PARP1 catalytically-dead alleles and/or analysis of the effect of PARP1 specific inhibitors.

2. The Timeless-binding site of human DDX11 has been recently identified and DDX11 site-specific mutants (named DDX11 KAE and KAK) have been described that are unable to directly interact with Timeless [Cortone et al., PLoS Genetics, 2018]. I suggest that complementation assays are carried out with these DDX11 Timeless-binding defective mutants. If the results of this analysis will reveal that the DDX11/Timeless interaction is critical for G-quadruplex handling at the replication forks, this finding would better substantiate the hypothesis that Timeless promotes resolution of these structures by recruiting and activating the DDX11 DNA helicase.

3. It has not yet been established if DDX11 is a "constitutive" component of the replisome in DT40 chicken cells or if it associates to the replication forks only when they are stalled, as it happens at sites where stable G-quadruplex structures are present. The Authors should clarify this point by carrying out iPOND analyses [Sirbu et al., Nat. Protoc., 2012] and/or by using the SIRF technique [Roy et al., J. Cell Biol., 2018] to examine the association of DDX11 (and Timeless) at sites of DNA synthesis in unchallenged conditions and after treatment with Pyridostatin. Alternatively, co-immuno-precipitation experiments of the chromatin fraction from cells treated (or not) with the above G-quadruplex-binder should be carried out with an antibody directed against a stable replisome component (such as Cdc45 or AND-1 or Timeless) and the presence of DDX11 in the pulled down samples analysed by Western blot.

Minor points:

1. In the complementation assays reported in Figure 4, the Authors should examine the expression level of full-length Timeless and its truncated forms by Western blot analysis of the DT40 cell extracts to rule out the possibility that the observed phenotype may depend on reduced stability of the corresponding deleted forms of Timeless. Besides, the Authors should verify that association of the Timeless deletion mutants to the DNA replication forks is not reduced compared to the full-length protein.

2. Position of the numbers and the name "Timeless" should be corrected in the schematic drawing of the Timeless polypeptide chain of Figure 2 Panel A.

Referee #2:

In this paper, Lerner et al. show that Timeless, a component of the fork protection complex (FPC) that travels with the replisome, has a role in replicating past G4 DNA. This is shown by an in vivo genetic assay that can detect replication efficiency of the Bu-1 gene in chicken DT40 cells. They map a specific DNA binding domain within Timeless that contributes to this function, and determine its structure. Lastly, they show that this domain works together with the PARP binding domain as well as the DDX11 helicase, but in a separate pathway from the FANCD1 helicase, to facilitate G4 replication. Altogether, this study adds significant new information about the role of Timeless in replication and characterizes a new, specific function in replicating G4 DNA, which will be of broad interest to the field. The experiments are well-conducted and the combination of in vivo genetic data and domain analysis strengthens the conclusions.

The main issue is that a few of the experiments are not explained well or interpreted sufficiently to support the conclusions, as outlined below.

Major concern that should be addressed

Fig. 6B & C: How was H2AX phosphorylation detected by flow cytometry exactly? This should be explained in the text and legend, not just in the methods (where details are also missing). For example it should at least be stated that it was detected by an H2A antibody specific for the phosphorylated form with a fluorescent (? Not stated and needs to be) secondary antibody. Also, how much pH2A is necessary for a cell to be "positive"? What would this correlate to in terms of number of pH2A foci in the cell (0 vs 1?; 10s vs 100s)? A micrograph should be added that shows a few typical WT, timeless, and ddx11 cells treated with the same antibodies so that the reader can determine the level of foci that are typical for these conditions and what the flow experiment is actually measuring.

Minor concerns that should be addressed

Fig 7B is difficult to understand from the information in the figure legend and results text. What is the "cumulative fraction" (on figure) or "cumulative distribution of genes" (in legend)? It would seem most genes would have 10 or less G4 motifs within 1.5 kb of the TSS so the larger fraction of genes above that number is confusing. Figures should be explained well enough in the text and legend that the average reader can understand them without consulting the methods (anyways I couldn't find this information in the methods).

The interpretation of Fig. 7D is not explained clearly: "Further, the enrichment of G4s near the TSS was similar to that seen in cells lacking the FANCD1 helicase (Fig. 7B & 7D)"

The authors need to specify in the text (not just legend) that Fig. 7D shows genes that are up- or down-regulated in those backgrounds compared to wt. Also point out to the reader what in the altered pattern compared to WT indicates a greater enrichment of G4s - the peak above the black line before the TSS or after or both? And how do G4s before or after the TSS affect transcription - what is the significant of the difference in pattern compared to WT? Just saying the pattern is similar to FANCI is too little information for the reader to understand the significance of the result.

Referee #3:

In this manuscript Lerner LK et al describe Timeless C-terminal DNA-binding domain preferentially binds to G-quadruplex DNA. They demonstrate that this Timeless DBD and the adjacent PARP1-binding domain are required for processive replication of G4-forming DNA. The authors further claim that Timeless and DDX11 work together to process G4 DNA structures and maintain processive DNA synthesis.

1) The Timeless region encompassing residues 816-954 represents the DNA-binding domain. The authors applied X-ray crystallography and NMR spectroscopy to determine structures of N- and C-terminal fragments of Timeless DNA-binding domain (Figure 2). Unfortunately, these structures provide very limited information in terms of how exactly Timeless DBD binds ssDNA or dsDNA, not even to mention the molecular basis of G-quadruplex DNA recognition.

2) Two models proposed in Figure 7 are purely speculative. The authors propose that Timeless senses G4 structure and further recruits DDX11 to unwind G4 DNA. The authors did not provide any evidence to support that DDX11 is recruited to G4 DNA by Timeless right after the Timeless senses G-quadruplex (through its DBD). Previous studies have shown that DDX11 physically interacts with Timeless (PMID: 20124417; 26503245; 30303954), the authors also did co-IP to show two proteins coexist in the same complex (Figure EV3). Is it possible that DDX11 forms a constitutive complex with Timeless at replication fork during S and G2 phases? Also, can Timeless, Tipin and PARP1 coexist in the same complex? Young LM et al (PMID: 26456830) have previously shown that Timeless/Tipin and Timeless/PARP1 are mutually exclusive.

3) Timeless/Tipin complex is associated with the replisome. Previous studies have demonstrated that knockdown either Timeless or Tipin could slow down replication rate, and cause cells more sensitive to DNA damage reagents. It is not surprising to observe increased levels of Bu-1 expression instability (Figure 1), increased activation of DDR in presence of pyridostatin (Figure 6) in Timeless or Tipin deficient cells compared to the wild-type cells. The conclusion in this study is mainly based on the observations from Timeless or DDX11 knockdown vs WT cells, I am not convinced by these indirect cell-based observations though Timeless DBD may have the binding preference to G-quadruplex DNA. The data interpretation of Timeless KD in this study could be more complicated. The authors may consider using AND-1 and RPA32 as the control, and check the level of Bu-1 expression instability in AND-1 KD and RPA32 KD DT40 cells.

4) The novelty of the study is limited, especially Cali F et al (PMID: 26503245) have already provided the data suggesting Timeless and DDX11 act together to preserve replication fork progression in perturbed conditions. Although the authors claim Timeless DBD exhibits specific binding to G-quadruplex DNA, the mechanistic details are lacking. I doubt the structures presented in the study could enhance our understanding of implicated functions.

Taken together, I don't think this study is suitable for publication.

Other points:

- i) The authors should provide the data to show the efficiency of Timeless, DDX11 knockdown. Also, does knockdown of Timeless have an impact on DDX11 protein level? As the previous study (Cali F et al. PMID: 26503245) indicated that stability of Timeless and DDX11 is interdependent.
- ii) The authors applied fluorescence polarization to show that Timeless DBD binds to dsDNA with Kd of $\sim 0.8\mu\text{M}$, and binds to G4 (in the promoter of Myc?) with Kd of $\sim 0.12\mu\text{M}$ (Figure 2). Timeless/Tipin complex binds to ssG4 with Kd of $\sim 0.6\mu\text{M}$ (Figure 3), and with a weaker affinity toward 1XAV (in the promoter of Myc?). Under the same condition, Timeless DBD and Timeless/Tipin complex, which one shows the stronger binding to G4? Considering the size of several DNA fragments (> 5 kDa), fluorescence polarization may not provide the accurate measurement. The authors should consider other approaches such as ITC, Biacore.
- iii) The authors showed that Timeless C-terminal region is not required for DDX11 interaction (Figure EV3). Why don't the authors map the Timeless region responsible for DDX11 binding? Then the authors can perform similar complementation assay (as shown in Figure 4) using Timeless mutants that just lost the binding to DDX11.
- iv) The authors also claim functional redundancy between Timeless DBD and PARP1-binding domain that is independent of DDX11 recruitment. Is this claim consistent with the sentence in the abstract "We show that this domain is required, in conjunction with an adjacent PARP-binding domain, to maintain processive replication through G4-forming sequences by promoting recruitment of the helicase DDX11"? How exactly does Timeless promote the recruitment of DDX11 to unwind G4?

Lerner, Holzer et al. Timeless couples G quadruplex detection with processing by DDX11 during DNA replication
Response to reviewers' comments.

Black: Full referees' comments
Blue: Our responses

Referee #1:

This work, carried out in collaboration by the laboratories of Julian Sale and Luca Pellegrini, describes a previously unrecognised DNA-binding domain (DBD) of Timeless, a component of the replication fork-protection complex, that shows high affinity for G-quadruplex-forming DNA oligonucleotides as revealed by in vitro binding assays. The Authors have carried out a biochemical and structural analysis of the Timeless DBD and have examined the role of this domain in assisting smooth replication of a DT40 genomic locus that harbours a sequence with the potential of forming a G-quadruplex structure downstream the BU-1 gene promoter. In previous studies by the Sale's laboratory, prolonged stalling of replication forks at this G-quadruplex structure was demonstrated to cause an altered epigenetic mark transmission at this genomic site and a consequently reduced expression of the surface glycoprotein encoded by the BU-1 gene, a change that can be directly monitored by a flow cytometry-based assay. Besides, the Authors shows that Timeless acts in concert with the DDX11 DNA helicase in promoting replication through this G-quadruplex-containing DNA template, in agreement with a previous report that revealed a direct physical and functional interaction between Timeless and DDX11 in human cells during DNA replication in stressful conditions [Cali et al., 2016, NAR]. At the end of the manuscript, a model is proposed where Timeless senses and binds G-quadruplex structures that arise at the DNA replication forks and recruits DDX11 to promote resolution of these structures in a pathway found to be redundant to the one of the paralogous FANCD1 DNA helicase.

This study is extremely novel and significant and the reported findings are of paramount interest in the field of genome stability/chromosome dynamics, especially considering that the role of the mammalian fork-protection complex in assisting progression of the replication machinery through difficult-to-replicate DNA templates is not fully understood from a mechanistic standpoint. The results of this work and the derived conclusions are persuasive. Nonetheless, before publication in EMBO Journal, some additional experiments need to be carried out to reinforce conclusions and the proposed model about the roles played by Timeless, DDX11 and PARP1 when DNA replication forks are stalled at sites forming G-quadruplex structures, as specified below.

Major points:

1. The possible role of PARP1 in the resolution of G-quadruplex DNA structures at the replication forks needs to be further investigated. The Authors have found that the Timeless PARP1-binding domain (PBD) is required for a complete rescue of the BU-1 genetic instability in avian cells. I suggest that the Authors examine if PARP1 operates in this or a redundant pathway of G-quadruplex resolution and if the PARP1 catalytic activity is required for G-quadruplex processing. This would require the use of PARP1-KO DT40 cells already described [Ooka et al., PLoS One, 2018] and complementation assays with PARP1 catalytically-dead alleles and/or analysis of the effect of PARP1 specific inhibitors.

Further examination of the role of PARP1 in recognising G4s at the replication fork will be an interesting extension of the work. However, a proper exploration of this point goes beyond the scope of the current paper.

We believe our identification of a DNA binding domain in a core component of the replisome that preferentially recognises G4s is an important step forward in understanding the response of the replisome to secondary structure.

2. The Timeless-binding site of human DDX11 has been recently identified and DDX11 site-specific mutants (named DDX11 KAE and KAK) have been described that are unable to directly interact with Timeless [Cortone et al., PLoS Genetics, 2018]. I suggest that complementation assays are carried out with these DDX11 Timeless-binding defective mutants. If the results of this analysis will reveal that the DDX11/Timeless interaction is critical for G-quadruplex handling at the replication forks, this finding would better substantiate the hypothesis that Timeless promotes resolution of these structures by recruiting and activating the DDX11 DNA helicase.

This is an important point. We have now complemented the *ddx11* mutant with DDX11 carrying the KAK mutation, previously shown to interrupt the interaction with Timeless (Cortone *et al.*, 2018). This mutant DDX11 is expressed (new Appendix Figure S5), but fails to complement the instability of *BU-1* expression, supporting the importance of the interaction between DDX11 and Timeless in counteracting replication pausing at secondary structures. These data are incorporated into revised Figure 5B.

Figure 2 (new Appendix Figure S5). Western blot of whole cell extract to show expression of FLAG-DDX11 KAK in four *ddx11* clones. Wild type (WT) and WT cells transfected with full length, unmutated DDX11 are shown as a control. Loading is controlled by blotting for PCNA.

Figure 3. The DDX11-KAK mutant does not reverse the BU-1 expression instability of the ddx11 mutant DT40 cells. **** $p < 0.0001$ one-way ANOVA compared with ddx11. The results presented for the KAK mutant are pooled from the four clones in Figure 2. These data are incorporated into new Figure 5.

3. It has not yet been established if DDX11 is a "constitutive" component of the replisome in DT40 chicken cells or if it associates to the replication forks only when they are stalled, as it happens at sites where stable G-quadruplex structures are present. The Authors should clarify this point by carrying out iPOND analyses [Sirbu et al., Nat. Protoc., 2012] and/or by using the SIRF technique [Roy et al., J. Cell Biol., 2018] to examine the association of DDX11 (and Timeless) at sites of DNA synthesis in unchallenged conditions and after treatment with Pyridostatin. Alternatively, co-immuno-precipitation experiments of the chromatin fraction from cells treated (or not) with the above G-quadruplex-binder should be carried out with an antibody directed against a stable replisome component (such as Cdc45 or AND-1 or Timeless) and the presence of DDX11 in the pulled down samples analysed by Western blot.

It remains an interesting question as to whether DDX11 is a constitutive component of the replisome or whether it is recruited dynamically. DDX11 was not identified at unperturbed or HU-stressed forks in the original iPOND experiments of Cortez (Dungrawala et al., 2015). However, a very recent paper has reported that DDX11 interacts with AND-1 and POLD1 (Simon et al., 2020), both constitutive components of the replisome. DDX11 has also been reported to bind to PCNA (Farina et al., 2008). iPOND and SIRF have not yet been established in DT40 and would be severely limited by lack of effective antibodies for immunofluorescence. Indeed, availability of antibodies that work well on chicken proteins is a more general limitation when working with DT40. Nonetheless, we have adopted the second suggestion of the reviewer and performed ChIP with PCNA, which is a completely conserved at the amino acid level between chicken and human, probing for tagged DDX11 with and without incubation of the cells with PDS. In two independent experiments we observe that treatment with PDS increases DDX11 association with chromatin and the amount pulled down by PCNA. These data are now incorporated in Figure 5A (with the replicate in the Appendix Fig. S4)

Figure 4. ChIP with PCNA followed by detection of FLAG-DDX11 in complemented *ddx11* DT40 cells.

Summary of protocol: 1) Treat cells with 4 μ M PDS or equivalent DMSO for 24 h. 2) Crosslink with 1% formaldehyde. 3) Extract chromatin and shear to average ~500 bp. 4) Pull down with PCNA (PC10). 5. Wash stringently (low salt, high salt, LiCl wash and TBST). 6. Reverse crosslinking and elute by boiling with Laemmli buffer. 7) Load input (1%) and the eluates, blot with FLAG (for DDX11), tubulin or PCNA. We have included the C-20 blot here to show that we are pulling down PCNA, but have removed it from the main figure.

Figure 5. Repeat of PCNA-DDX11 ChIP +/- PDS. Method as in Figure 4.

Minor points:

1. In the complementation assays reported in Figure 4, the Authors should examine the expression level of full-length Timeless and its truncated forms by Western blot analysis of the DT40 cell extracts to rule out the possibility that the observed phenotype may depend on reduced stability of the corresponding deleted forms of Timeless. Besides, the Authors should verify that association of the Timeless deletion mutants to the DNA replication forks is not reduced compared to the full-length protein.

We have verified expression of each FLAG-tagged mutant and their association to chromatin. Note an endogenous protein in chicken, which we have identified by mass spectrometry as HSPH1 (Hsp110) contains a FLAG-like epitope at its C-terminus (L.G. Phillips University of Cambridge PhD thesis 2006):

HSPH1 810-KEENPLNEQGDKYKTEDMGEDDKNSDNPQQNGECHPGDQNTVNMDLD-856
 FLAG DYK----DDDDK
 *** .:***

Figure 6. The FLAG-like epitope in chicken HSPH1. This protein has both cytoplasmic and nuclear localisation and interacts with the transcription factors c-Myc and BCL-6 in human lymphoma cells (Zappasodi *et al.*, 2015).

We are able to detect expression of each of the mutants in both the whole cell extract and associated with chromatin. The Δ CTD mutant protein may be unstable or more poorly expressed, but is still associates with chromatin at levels comparable to the other wild type and other mutants.

Figure 7. Detection of FLAG-Timeless and mutants. A. Anti-FLAG blot on whole cell extract and B. chromatin associated proteins (prepared as outlined in the legend to Fig 4). Incorporated as new Appendix Fig. S3.

2. Position of the numbers and the name "Timeless" should be corrected in the schematic drawing of the Timeless polypeptide chain of Figure 2 Panel A.

Corrected.

Referee #2:

In this paper, Lerner *et al.* show that Timeless, a component of the fork protection complex (FPC) that travels with the replisome, has a role in replicating past G4 DNA. This is shown by an *in vivo* genetic assay that can detect replication efficiency of the Bu-1 gene in chicken DT40 cells. They map a specific DNA binding domain within Timeless that contributes to this function, and determine its structure. Lastly, they show that this domain works together with the PARP binding domain as well as the DDX11 helicase, but in a separate pathway from the FANCD1 helicase, to facilitate G4 replication. Altogether, this study adds significant new information about the role of Timeless in replication and characterizes a new, specific function in replicating G4 DNA, which will be of broad interest to the field. The experiments are well-conducted and the combination of *in vivo* genetic data and domain analysis strengthens the conclusions.

The main issue is that a few of the experiments are not explained well or interpreted sufficiently to support the

conclusions, as outlined below.

Major concern that should be addressed

Fig. 6B & C: How was H2AX phosphorylation detected by flow cytometry exactly? This should be explained in the text and legend, not just in the methods (where details are also missing). For example it should at least be stated that it was detected by an H2A antibody specific for the phosphorylated form with a fluorescent (? Not stated and needs to be) secondary antibody. Also, how much pH2A is necessary for a cell to be "positive"? What would this correlate to in terms of number of pH2A foci in the cell (0 vs 1?; 10s vs 100s)? A micrograph should be added that shows a few typical WT, *timeless*, and *ddx11* cells treated with the same antibodies so that the reader can determine the level of foci that are typical for these conditions and what the flow experiment is actually measuring.

We have improved and clarified the description of this experiment. As requested, we have also now provided IF images of cells prepared and stained for γ H2AX in exactly the same manner as for the cytometry experiments to demonstrate exactly what the cytometer is measuring. Further, we have quantitated foci of γ H2AX using FIJI to find and count foci. Obviously, there is a limit to the quantification of foci when the signal is extensive as the foci merge resulting in a pan-nuclear signal. Nonetheless, the two methods, cytometry and microscopy, give similar results. The images and quantitation of foci have been added as Appendix Fig. S6.

Figure 8. Examples of γ H2AX immunofluorescence and quantitation of γ H2AX foci in wild type, *ddx11* and *timeless* DT40 cells with and without exposure to PDS at 4 μ M for 24 hours. The cells were treated and prepared exactly as for cytometry experiments in Figure 6.

Minor concerns that should be addressed

Fig 7B is difficult to understand from the information in the figure legend and results text. What is the "cumulative fraction" (on figure) or "cumulative distribution of genes" (in legend)? It would seem most genes would have 10 or less G4 motifs within 1.5 kb of the TSS so the larger fraction of genes above that number is confusing. Figures

should be explained well enough in the text and legend that the average reader can understand them without consulting the methods (anyways I couldn't find this information in the methods).

The interpretation of Fig. 7D is not explained clearly: "Further, the enrichment of G4s near the TSS was similar to that seen in cells lacking the FANCI helicase (Fig. 7B & 7D)"

The authors need to specify in the text (not just legend) that Fig. 7D shows genes that are up- or down-regulated in those backgrounds compared to wt. Also point out to the reader what in the altered pattern compared to WT indicates a greater enrichment of G4s - the peak above the black line before the TSS or after or both? And how do G4s before or after the TSS affect transcription - what is the significance of the difference in pattern compared to WT? Just saying the pattern is similar to FANCI is too little information for the reader to understand the significance of the result.

We have improved the presentation of this data in the text, legend and methods and endeavoured to ensure that the figures and legends can be understood in isolation.

Referee #3:

In this manuscript Lerner LK et al describe Timeless C-terminal DNA-binding domain preferentially binds to G-quadruplex DNA. They demonstrate that this Timeless DBD and the adjacent PARP1-binding domain are required for processive replication of G4-forming DNA. The authors further claim that Timeless and DDX11 work together to process G4 DNA structures and maintain processive DNA synthesis.

1) The Timeless region encompassing residues 816-954 represents the DNA-binding domain. The authors applied X-ray crystallography and NMR spectroscopy to determine structures of N- and C-terminal fragments of Timeless DNA-binding domain (Figure 2). Unfortunately, these structures provide very limited information in terms of how exactly Timeless DBD binds ssDNA or dsDNA, not even to mention the molecular basis of G-quadruplex DNA recognition.

Our paper reports the first identification of a DNA-binding domain in Timeless, together with the biochemical characterization of its DNA-binding properties. This is an important advance that sheds new light on to how the replisome responds to DNA secondary structure. We agree with the reviewer that understanding the structural basis for specific recognition of a G4 DNA by the Tim-DBD would be highly desirable, but that such a study is beyond the scope of this manuscript.

2) Two models proposed in Figure 7 are purely speculative. The authors propose that Timeless senses G4 structure and further recruits DDX11 to unwind G4 DNA. The authors did not provide any evidence to support that DDX11 is recruited to G4 DNA by Timeless right after the Timeless senses G-quadruplex (through its DBD). Previous studies have shown that DDX11 physically interacts with Timeless (PMID: 20124417; 26503245; 30303954), the authors also did co-IP to show two proteins coexist in the same complex (Figure EV3). Is it possible that DDX11 forms a constitutive complex with Timeless at replication fork during S and G2 phases?

The models proposed in Figure 7 are indeed speculative and is, like any model, simply intended to provide guidance as to our interpretation and discussion of the data we present. As discussed above, we agree that we perhaps overplayed the idea of sequential recruitment of Timeless then DDX11 to a G4. Our genetic data are consistent with either a recruitment model with or a constitutive complex and these interpretations and whether the complex is constitutive or not does not alter our conclusions. However, the ChIP data that we have now added in the new Figure 5A, and discussed above, shows that PDS exposure increases the amount of DDX11 bound to PCNA and chromatin, suggesting that there is indeed an inducible element to DDX11 recruitment. We have revised our discussion and model accordingly.

Also, can Timeless, Tipin and PARP1 coexist in the same complex? Young LM et al (PMID: 26456830) have previously shown that Timeless/Tipin and Timeless/PARP1 are mutually exclusive.

We note that the findings of Young et al. (Young *et al.*, 2015) are not immediately consistent with findings of Xie et al (Mol Cell 2015). The latter paper demonstrates that Timeless recruitment to DNA lesions is dependent on PARP-1 (not the PARylation activity of the enzyme) and that Tipin recruitment also depends on the interaction between PARP-1 and Timeless. The interactions between Tipin, Timeless and PARP-1 require further investigation, particularly with respect G4 binding, but this is not within the scope of the current study and would require us to be able to detect Tipin in DT40 cells, which we currently cannot do. While we do not feel that this question affects our conclusions, we have added a discussion of this point to the paper.

3) Timeless/Tipin complex is associated with the replisome. Previous studies have demonstrated that knockdown

either Timeless or Tipin could slow down replication rate, and cause cells more sensitive to DNA damage reagents. It is not surprising to observe increased levels of Bu-1 expression instability (Figure 1), increased activation of DDR in presence of pyridostatin (Figure 6) in Timeless or Tipin deficient cells compared to the wild-type cells. The conclusion in this study is mainly based on the observations from Timeless or DDX11 knockdown vs WT cells, I am not convinced by these indirect cell-based observations though Timeless DBD may have the binding preference to G-quadruplex DNA. The data interpretation of Timeless KD in this study could be more complicated. The authors may consider using AND-1 and RPA32 as the control, and check the level of Bu-1 expression instability in AND-1 KD and RPA32 KD DT40 cells.

We would first like to clarify that none of the experiments presented in the manuscript employ knockdowns. All mutants are genetic disruptions of the respective genes. It is not possible to generate knockouts of AND-1 or RPA as these are essential and therefore not amenable to use in our *BU-1* assay.

We disagree that the slowing of replication caused by loss of Timeless would per se be expected to cause *BU-1* instability. We demonstrate that the observed instability is entirely dependent on the +3.5 G4 within the *BU-1* locus and is not therefore a general effect of replication slowing.

4) The novelty of the study is limited, especially Cali F et al (PMID: 26503245) have already provided the data suggesting Timeless and DDX11 act together to preserve replication fork progression in perturbed conditions. Although the authors claim Timeless DBD exhibits specific binding to G-quadruplex DNA, the mechanistic details are lacking. I doubt the structures presented in the study could enhance our understanding of implicated functions.

We naturally disagree that the identification of a specific DNA binding domain with preference for an important DNA secondary structure within a core component of the replisome is of limited novelty. We do, however, concur with the referee that more work is required on the precise nature of the structural interaction of the Timeless DBD with G4 DNA, but this is beyond the scope of the current manuscript.

Taken together, I don't think this study is suitable for publication.

Other points:

i) The authors should provide the data to show the efficiency of Timeless, DDX11 knockdown. Also, does knockdown of Timeless have an impact on DDX11 protein level? As the previous study (Cali F et al. PMID: 26503245) indicated that stability of Timeless and DDX11 is interdependent.

As noted above, these are not knockdown experiments, but genetic disruptions. We show the nature of the deletions introduced (Appendix Figure S1) and confirm that the phenotypes we observe are reversed by complementation. We cannot directly confirm the expression of endogenous DDX11 and Timeless in DT40 as the human antibodies do not work sufficiently well on the chicken proteins.

ii) The authors applied fluorescence polarization to show that Timeless DBD binds to dsDNA with K_d of $\sim 0.8\mu\text{M}$, and binds to G4 (in the promoter of *Myc*?) with K_d of $\sim 0.12\mu\text{M}$ (Figure 2). Timeless/Tipin complex binds to ssG4 with K_d of $\sim 0.6\mu\text{M}$ (Figure 3), and with a weaker affinity toward 1XAV (in the promoter of *Myc*?). Under the same condition, Timeless DBD and Timeless/Tipin complex, which one shows the stronger binding to G4? Considering the size of several DNA fragments (> 5 kDa), fluorescence polarization may not provide the accurate measurement. The authors should consider other approaches such as ITC, Biacore.

The affinity constants measured for Timeless DBD and Timeless-Tipin binding to G4 DNA in Figs. 2 and 3 respectively should not be directly compared, as the DNA substrates differ. In the DNA binding data of Fig. 3A, the G4 was embedded in a longer ssDNA sequence, to mimic the unwound DNA template before nucleotide polymerisation. In general, we find that Timeless DBD binds more tightly to G4 DNA than Timeless-Tipin, although we have not explored this issue extensively. It is possible that a fraction of our Timeless-Tipin sample might be inactive due to aggregation or misfolding, or that the DBD might partially be masked in the complex, which would point to the existence of a regulatory mechanism for DNA binding. What is important and relevant to the conclusions of our current study is that we see a clear preference for G4 DNA over ss- and dsDNA substrates for both Timeless DBD and the Timeless-Tipin complex.

We respectfully disagree with the reviewer's comment that fluorescent polarization would not provide accurate measurements of protein - DNA affinities. FP is widely accepted as one for the standard techniques for affinity measurements between proteins and oligonucleotides, as it provides fast and accurate information. The sizes of the DNA substrates used — between 7 and 18kDa — are well within range for FP measurements. Furthermore,

we would like to point out that we have also used EMSA assays to measure the relative binding affinity of Timeless-Tipin for DNA (Figure EV2).

iii) The authors showed that Timeless C-terminal region is not required for DDX11 interaction (Figure EV3). Why don't the authors map the Timeless region responsible for DDX11 binding? Then the authors can perform similar complementation assay (as shown in Figure 4) using Timeless mutants that just lost the binding to DDX11.

We have not identified the region of Timeless necessary for the interaction with DDX11 beyond showing that it is not in the C terminus of Timeless and further dissection of this interaction from the point of view of Timeless is beyond the scope of the present work. However, we now provide clear evidence that the previously identified domain in DDX11 that mediates its interaction with Timeless is central to the maintenance of *BU-1* expression stability and therefore for effective maintenance of processive replication through the +3.5 G4 in the locus.

iv) The authors also claim functional redundancy between Timeless DBD and PARP1-binding domain that is independent of DDX11 recruitment. Is this claim consistent with the sentence in the abstract "We show that this domain is required, in conjunction with an adjacent PARP-binding domain, to maintain processive replication through G4-forming sequences by promoting recruitment of the helicase DDX11"? How exactly does Timeless promote the recruitment of DDX11 to unwind G4?

On reflection, we agree that this wording here is not quite right. The apparent functional redundancy between the DBD and PBD is not necessarily independent of DDX11, although these domains are not responsible for DDX11 binding. Neither expression of the DNA- nor PARP1-binding mutant fully complements the *BU-1* instability of the *timeless* mutant. Nonetheless, they reduce it significantly more than the mutant lacking the entire C terminus encompassing both domains. This suggests that both domains play a role in suppressing *BU-1* expression instability, but that they are also to some extent able to cover for the absence of the other. A possible explanation for this would be that direct binding of Timeless to a G4 or indirect recruitment via PARP1 are able to, at least in part, compensate for each other in allowing the Timeless-DDX11 complex to operate efficiently in promoting processive replication through the structure. We have clarified our discussion of these results and, as discussed above, we have clarified the discussion of our proposed model for the recruitment and action of Timeless and DDX11 at G4s encountered during replication.

References

- Cogoi S, Paramasivam M, Membrino A, Yokoyama KK, Xodo LE (2010) The KRAS promoter responds to Myc-associated zinc finger and poly(ADP-ribose) polymerase 1 proteins, which recognize a critical quadruplex-forming GA-element. *J Biol Chem*, **285**: 22003–22016
- Cortone G, Zheng G, Pensieri P, Chiappetta V, Tatè R, Malacaria E, Pichierri P, Yu H, Pisani FM (2018) Interaction of the Warsaw breakage syndrome DNA helicase DDX11 with the replication fork-protection factor Timeless promotes sister chromatid cohesion. *PLoS Genet*, **14**: e1007622
- Dungrawala H, Rose KL, Bhat KP, Mohni KN, Glick GG, Couch FB, Cortez D (2015) The Replication Checkpoint Prevents Two Types of Fork Collapse without Regulating Replisome Stability. *Mol Cell*, **59**: 998–1010
- Farina A, Shin JH, Kim DH, Bermudez VP, Kelman Z, Seo YS, Hurwitz J (2008) Studies with the human cohesin establishment factor, ChIR1. Association of ChIR1 with Ctf18-RFC and Fen1. *J Biol Chem*, **283**: 20925–20936
- Simon AK, Kummer S, Wild S, Lezaja A, Teloni F, Jozwiakowski SK, Altmeyer M, Gari K (2020) The iron-sulfur helicase DDX11 promotes the generation of single-stranded DNA for CHK1 activation. *Life Sci Alliance*, **3**: e201900547
- Soldatenkov VA, Vetcher AA, Duka T, Ladame S (2008) First evidence of a functional interaction between DNA quadruplexes and poly(ADP-ribose) polymerase-1. *ACS Chem Biol*, **3**: 214–219
- Young LM, Marzio A, Perez-Duran P, Reid DA, Meredith DN, Roberti D, Star A, Rothenberg E, Ueberheide B, Pagano M (2015) TIMELESS Forms a Complex with PARP1 Distinct from Its Complex with TIPIN and Plays a Role in the DNA Damage Response. *Cell Rep*, **13**: 451–459
- Zappasodi R, Ruggiero G, Guarnotta C, Tortoreto M, Tringali C, Cavanè A, Cabras AD, Castagnoli L, Venerando B, Zaffaroni N, Gianni AM, De Braud F, Tripodo C, Pupa SM, Di Nicola M (2015) HSPH1 inhibition downregulates Bcl-6 and c-Myc and hampers the growth of human aggressive B-cell non-Hodgkin lymphoma. *Blood*, **125**: 1768–1771

Dr. Julian E. Sale
MRC Laboratory of Molecular Biology
Division of Protein & Nucleic Acid Chemistry
Francis Crick Avenue
Cambridge Biomedical Campus
Cambridge CB2 0QH
United Kingdom

24th Jun 2020

Re: EMBOJ-2019-104185R

Timeless couples G-quadruplex detection with processing by DDX11 during DNA replication

Thank you for submitting your revised manuscript to The EMBO Journal. It has now been assessed once more by two of the original referees, both of whom are fully satisfied with the revisions. I am therefore happy to inform you that we would like to proceed further with publication of this work, following final modifications to incorporate various editorial points as detailed below.

REFEREE REPORTS

Referee #1:

This manuscript revised version deserves publication in EMBO Journal: the experiments I suggested have been carried out and incorporated into it.

Referee #2:

Review of Lerner..Sale Resubmission (Reviewer 2):

The authors have addressed all my concerns.

They have also added important new data requested by reviewer 1 utilizing a DDX11 mutant that can no longer interact with Timeless. This mutant fails to complement the instability of BU-1 expression, supporting the importance of the interaction between DDX11 and Timeless in counteracting replication pausing at secondary structures. They also now show that treatment with PDS increases DDX11 association with chromatin and the amount pulled down by PCNA. They have verified expression levels of full-length Timeless and its truncated forms and determined their chromatin association by Western blot analysis. This new data strengthens the manuscript. They have also improved some wording issues related to the model requested by reviewer 3. I disagree with reviewer 3's conclusion that the novelty of this study is limited. I think that how the replisome senses and deals with secondary structures has been a mystery, and this paper provides an important first step in understanding that process.

Corresponding Author Name: Julian Sale & Luca Pellegrini

Journal Submitted to: EMBO J

Manuscript Number: EMBOJ-2019-104185